# Bipartite binding interface recruiting HP1 to chromosomal passenger complex at inner centromeres

Kosuke Sako[1], Ayako Furukawa[2,3], Ryu-Suke Nozawa[1], Jun-ichi Kurita[2], Yoshifumi Nishimura[2], and Toru Hirota[1]

**Maintenance of ploidy depends on the mitotic kinase Aurora B, the catalytic subunit of the chromosomal passenger complex (CPC) whose proficient activity is supported by HP1 enriched at inner centromeres. HP1 is known to associate with INCENP of the CPC in a manner that depends on the PVI motif conserved across HP1 interactors. Here, we found that the interaction of INCENP with HP1 requires not only the PVI motif but also its C-terminally juxtaposed domain. Remarkably, these domains conditionally fold the β-strand (PVI motif) and the α-helix from a disordered sequence upon HP1 binding and render INCENP with high affinity to HP1. This bipartite binding domain termed SSH domain (Structure composed of Strand and Helix) is necessary and sufficient to attain a predominant interaction of HP1 with INCENP. These results identify a unique HP1-binding module in INCENP that ensures enrichment of HP1 at inner centromeres, Aurora B activity, and thereby mitotic fidelity.**

## Introduction

Errors in chromosome segregation cause aneuploidy, a well-known feature of malignant tumors (Gordon et al., 2012; Santaguida and Amon, 2015; Thompson et al., 2010). Accurate chromosome segregation depends on proper kinetochore-microtubule (KT-MT) attachments (Godek et al., 2015), which is established through the destabilization of incorrect error-prone attachments (Cimini et al., 2001; Thompson and Compton, 2011). The Aurora B kinase plays a central role in destabilizing these KT-MT mal-attachments by phosphorylation of kinetochore proteins mediating MT attachments (Cimini et al., 2006), such as Hec1 (the Ndc80 complex), Dsn1 (the Mis12 complex), and Knl1 (Cheeseman et al., 2006; DeLuca et al., 2006, 2011; Ciferri et al., 2008; Welburn et al., 2010).

The chromosomal passenger complex (CPC) comprises INCENP, Survivin, Borealin/Dasra, and Aurora B. INCENP serves as a scaffold that accommodates the other components to regulate the localization and the activity of Aurora B: Survivin and Borealin/Dasra associate to the amino-terminus of INCENP, thereby interacting with phosphorylated histone tails to promote chromosomal association of the CPC (Jeyaprakash et al., 2007; Kelly et al., 2010; Wang et al., 2010; Yamagishi et al., 2010). Aurora B interacts with its carboxyl terminus through a domain termed IN-box. This interaction promotes an allosteric activation of Aurora B, leading to autophosphorylation of the catalytic activation loop of Aurora B and the TSS motif in the IN-box (Bishop and Schumacher, 2002; Honda et al., 2003; Kelly et al., 2007; Sessa et al., 2005). These two functional modules are connected by a long intervening unstructured, intrinsically disordered region and a single alpha helix called the SAH domain (Samejima et al., 2015). Within the former, heterochromatin protein 1 (HP1) is known to directly bind to INCENP as an accessory subunit of the CPC (Abe et al., 2016; Ainsztein et al., 1998; Kang et al., 2011; Nozawa et al., 2010).

In interphase, HP1 is typically localized at heterochromatin, mediated by the binding between its amino-terminal, chromodomain (CD), and di-/trimethylated histone H3 lysine 9 (Bannister et al., 2001; Lachner et al., 2001; Li et al., 2002; Nielsen et al., 2001). In mitosis, most of the HP1 dissociate from chromatin and diffuse into the cytoplasm (Fischle et al., 2005; Hirota et al., 2005), except for a small fraction enriched at inner centromeres (Abe et al., 2016; Hayakawa et al., 2003; Serrano et al., 2009). This inner centromeric-localization of HP1 seems to depend on INCENP because the expression of HP1 binding-region deficient mutant of INCENP failed to enrich HP1 at centromeres (Abe et al., 2016; Kang et al., 2011). The interaction between HP1 and INCENP is mediated through the C-terminal chromo-shadow domain (CSD) of HP1 and the characteristic PxVxL/I sequence (PVI motif) that lies within the unstructured domain of INCENP (Nozawa et al., 2010; Smothers and Henikoff, 2000; Thiru et al., 2004). It is notable that the

........................................................................................................................................................................

[1]Division of Experimental Pathology, Cancer Institute of the Japanese Foundation for Cancer Research, Tokyo, Japan; [2]Graduate School of Medical Life Science, Yokohama City University, Yokohama, Japan; [3]Graduate School of Agriculture, Kyoto University, Kyoto, Japan.

Correspondence to Toru Hirota: thirota@jfcr.or.jp; Yoshifumi Nishimura: nisimura@yokohama-cu.ac.jp.

inner centromeric HP1 and its binding to INCENP is required for a proficient error-correcting activity of Aurora B (Abe et al., 2016).

There are other centromere-enriched proteins with the PVI motif (Kang et al., 2011; Kiyomitsu et al., 2010; Nozawa et al., 2010; Yamagishi et al., 2008), such as Sgo1, a protein that preserves centromeric cohesion (Kang et al., 2011; Yamagishi et al., 2008). Unlike INCENP, the interaction of Sgo1 with HP1 is dispensable for its localization in mitosis (Kang et al., 2011; Serrano et al., 2009). If HP1 predominantly binds to INCENP over the other PVI proteins in centromeres, what might specify this interaction? The fact that INCENP's PVI motif resides within the intrinsically disordered region limits us to obtain solid details of HP1/INCENP interaction based on crystal structure analysis (Krenn and Musacchio, 2015). Here, we found that INCENP includes an additional HP1 binding site C-terminally next to the PVI motif. The NMR analysis indicated that the PVI motif adopts the β-strand and the additional HP1 binding site adopts the α-helix upon binding to HP1. These sites collectively called SSH domain (Structure composed of β-Strand and α-Helix) are characteristic of INCENP supporting the preferential binding to HP1. INCENP having an incompetent SSH domain reduced the amount of HP1 binding and caused an abnormal chromosome segregation behavior during anaphase. These results suggest that the bipartite HP1-binding domain of INCENP ensures CPC is the prime adaptor for HP1 at inner centromeres in mitosis.

## Results

### INCENP recruits HP1 at inner centromeres in mitosis

Provided that mitotic enrichment of HP1α at inner centromeres depends on INCENP (Abe et al., 2016; Kang et al., 2011), we expected to see a similar pattern of their behavior at inner centromeres. In fluorescence microscopy of spread chromosomes, in which their spatial distribution could be readily examined, enrichment of both HP1α and INCENP was detected as a single fluorescence peak at inner centromeres throughout prometaphase (Fig. 1 A and Fig. S1, B–D). In contrast, Sgo1 revealed a broader distribution at centromeres (Fig. 1 B and Fig. S1 E), as previously described (Liu et al., 2015). Thus, the localization of HP1α largely overlapped with that of INCENP rather than with Sgo1.

In agreement with these observations, RNAi-mediated depletion of Sgo1 hardly affected the inner centromeric enrichment of neither HP1α (Fig. 1, A and B) nor INCENP (Fig. 1, A and C). By contrast, depletion of INCENP caused a major displacement of HP1α from inner centromeres (Fig. 1, A and B), while the localization of Sgo1 could still be detected at centromeres (Fig. 1, B and C). Depletion of HP1α did not affect the localization of both INCENP and Sgo1 (Fig. S1 F), as previously shown (Kang et al., 2011; Serrano et al., 2009). These results are consistent with the idea that the primary receptor of HP1 at inner centromeres is INCENP in mitosis (Fig. 1 D).

In cells depleted of INCENP, HP1α remained distributed throughout chromosomes (Fig. 1, A and B), which led us to address the association of INCENP with HP1 before mitosis, given their interaction is known to presumably occur from the G2

phase (Nozawa et al., 2010). We conducted an immunoprecipitation of INCENP from G2– or M phase–enriched cell lysates and found that it precipitated with HP1 also in the G2 phase (Fig. 1 E). In this assay, HP1 coimmunoprecipitated with Sgo1 was hardly detectable in both G2 and M phase lysates. It became discernible in a lower salt condition, yet the amount was much smaller than that with INCENP (Fig. S1, G and H). These binding assays indicated a preponderance of HP1 bound INCENP in both the G2 and M phase in a more stable manner than it is bound to Sgo1, further underscoring the role of INCENP in recruiting HP1 at inner centromeres.

### PVI motif and its C-terminally juxtaposed domain are involved in the HP1 interaction

INCENP is predicted to have a long unstructured sequence, i.e., intrinsically disordered region (Krenn and Musacchio, 2015), which includes the HP1 binding domain (Fig. 2 A; and Fig. S2, A and B; Abe et al., 2016; Kang et al., 2011; Nozawa et al., 2010). The intrinsically disordered region contains the sequence of Pro167, Val169, and Ile171 known as the PVI motif, the canonical HP1 binding motif. To define the binding domain of INCENP to HP1, we first carried out an *in vitro* pull-down assay using purified recombinant proteins of human HP1α and INCENP (Fig. 2 B and Fig. S2 C). We found that INCENP fragments consisting of 121–270 amino acids (aa) and 160–210 aa bound to the HP1α, whereas the 121–178 fragment and the alanine mutant of PVI motif (Pro167Ala, Val169Ala, and Ile171Ala; PVI_3A) of 121–270 fragment significantly reduced the binding ability (Fig. 2 B). These data indicate that the PVI motif and its downstream sequence are both required to support a stable interaction with HP1.

To verify these results, we conducted an isothermal titration calorimetry (ITC) analysis in which the release or absorption of heat is measured as a readout of molecular interactions. ITC experiments quantitatively showed that INCENP 121–270, 121–210, and 160–210 fragments interacted specifically with HP1α, whereas 121–178 fragment and a short peptide spanning 159–178 failed to interact with HP1α despite the presence of the PVI motif (Fig. 2, C and D; and Fig. S2, C–F). This is in contrast to the binding of Sgo1 with HP1, as an equivalent fragment of Sgo1 including the PVI motif is known to interact with HP1 in an ITC analysis and a cocrystal structure (Kang et al., 2011).

These binding assays indicate that the HP1 binding domain of INCENP lies within a region of 160–210 amino acids and the PVI motif alone is not sufficient to bind to HP1α; an extension toward the C-terminal side from the PVI motif supports the specific interaction of INCENP with HP1.

### The disordered region adjacent to the PVI motif folds into α-helix upon HP1 binding

To examine the complex structure of the INCENP fragment of 160–210 residues bound to HP1α, we conducted nuclear magnetic resonance (NMR) spectroscopy using $^{15}$N- and $^{13}$C/$^{15}$N-labeled recombinant fragments (Fig. S3 A). The [$^{1}$H,$^{15}$N]-heteronuclear single-quantum coherence spectroscopy (HSQC) spectrum of the fragment revealed signals characteristic for disordered structure, i.e., clustered in $^{1}$H- and dispersed in $^{15}$N-chemical shifts (Fig. 3 A; Schwarzinger et al., 2001;

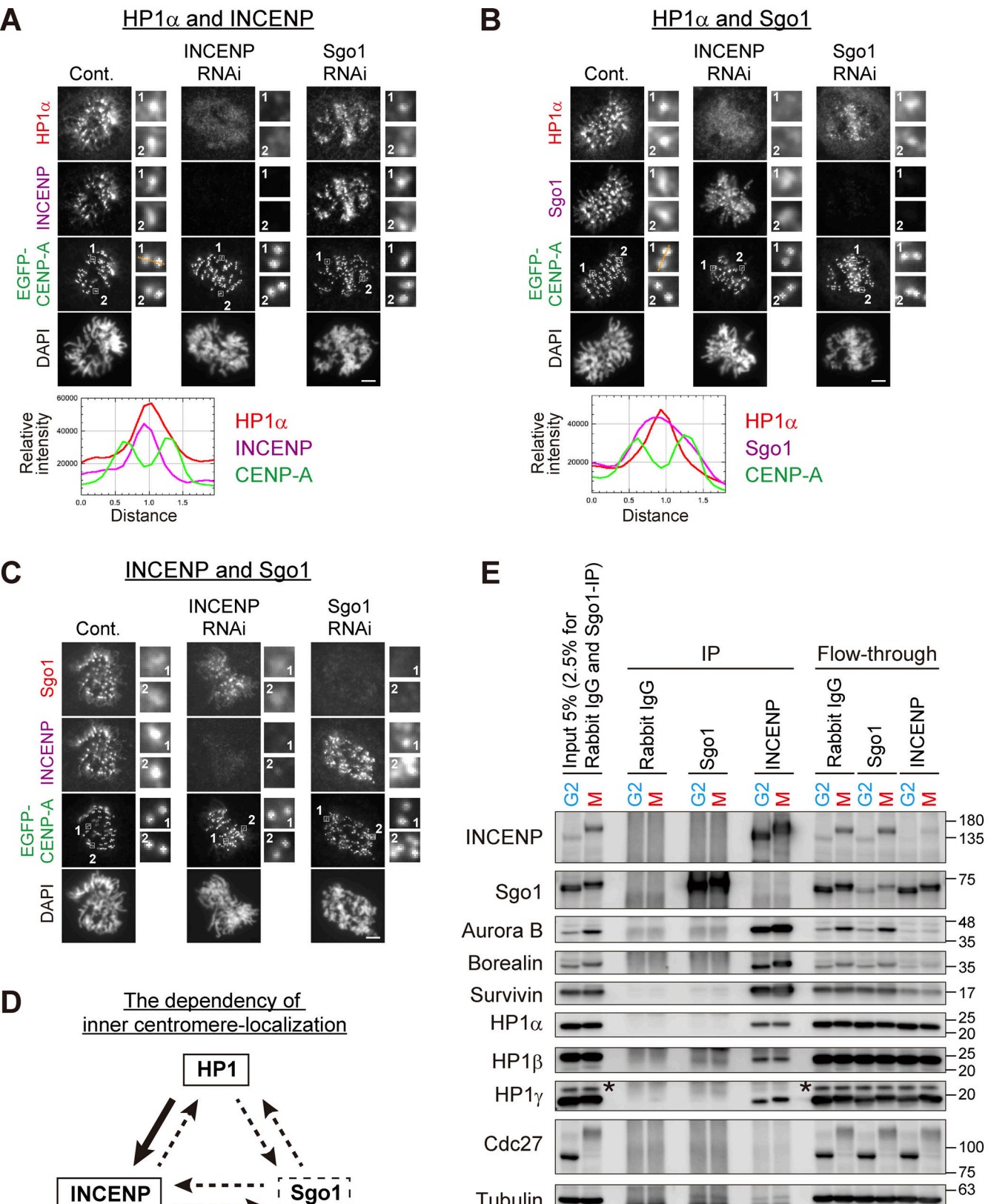

Figure 1. **Inner centromeric localization of HP1α depends on INCENP, not on Sgo1. (A–C)** Immunofluorescence microscopy of chromosome spreads from HeLa cells constitutively expressing EGFP-CENP-A. Cells transfected with siRNA to INCENP or Sgo1 during 24 h of thymidine block were fixed at 9.5 h after the release. The protein levels were verified by immunoblots as shown in Fig. S1 A. Cells were spun on glass slides (see Materials and methods for details) and the spread chromosomes were stained with antibodies to HP1α and INCENP (A), HP1α and Sgo1 (B), and Sgo1 and INCENP (C). A representative pattern of inner centromeric-localized proteins in field #1 is indicated by line scans for HP1α (red), INCENP or Sgo1 (magenta), and EGFP-CENP-A (green) (A and B). Scale bar,

5 µm. **(D)** Dependency of inner centromere-localization of HP1, INCENP, and Sgo1 during mitosis. The localization of HP1 depends on INCENP as indicated by the solid-line arrow. Dotted-line arrows indicate little or no dependency. **(E)** Predominant binding of HP1 to INCENP in G2 and M phase-enriched cell extracts. Synchronous HeLa cell populations in G2 or in M phase following double thymidine block and release were prepared by 10 µM RO3306 treatment for 9 h or 7.5 µM STLC treatment for 16 h, respectively, and cell extracts were subjected to immunoprecipitation (IP) assays. The synchronicity was verified by the electrophoresis velocity of Cdc27. Asterisks indicate non-specific proteins. In a condition where the majority of Sgo1 and INCENP in extracts were immunoprecipitated, HP1 was readily detected in INCENP immunoprecipitates but not in Sgo1, even though double the amount of Sgo1-IP samples was loaded. Source data are available for this figure: SourceData F1.

---

Schwarzinger et al., 2000). To delineate the binding region of HP1α, we performed NMR titration experiments in which unlabeled full-length HP1α and CSD of HP1α were titrated into $^{15}$N-labeled INCENP 160–210 (Fig. 3 B and Fig. S3 B). Consistent with the possibility that CSD is sufficient for HP1α to interact with INCENP, we found that both the full-length and CSD of HP1α similarly reduced signal intensities of the fragment, measurably in the region spanning 160–192 amino acids (Fig. 3 C). We then prepared a fragment of INCENP 160–192 aa and verified that it has a binding affinity to CSD equivalent to that of 160–210 aa (Fig. 3, D and E). These data suggested that INCENP His160-Leu192 is sufficient to bind to HP1α.

To characterize the binding interface, we therefore prepared INCENP 160–192 aa, $^{13}$C/$^{15}$N-labeled fragment (Fig. S3 C), and [$^{1}$H,$^{15}$N] HSQC spectra were analyzed in the absence or presence of HP1α CSD (Fig. 3, F and G). Notably, the use of this shorter INCENP fragment resulted in highly resolved signals, most of which were successfully assigned. To predict the secondary structure of the binding interface, we calculated the chemical shift indices based on the chemical shift value of $^{13}$C$_\alpha$ (Fig. 3 H and Fig. S3 D). The sequence around the PVI motif (164–169 aa) was found to adopt a β-strand conformation as expected; surprisingly, its C-terminal adjacent region (173–185 aa) folded into an α-helix upon HP1 binding. In mitotic cells, an INCENP mutant involving this adjacent region failed to stably bind to HP1α despite the presence of the PVI motif, and the amount of HP1α binding dropped to the level of PVI_3A mutant (Fig. S4 A). These data indicated that the binding of INCENP to HP1 depends on both the PVI motif (β-strand) and the adjacent α-helix, and neither is sufficient to support a stable binding by itself.

### HP1α dimer interacts with INCENP asymmetrically

Given the requirement of the induced α-helix to interact with HP1, we wish to know where in the CSD dimer HP1α mediates the binding. To address this, we prepared $^{13}$C/$^{15}$N-labeled HP1α CSD dimer and performed NMR titration experiments in which the unlabeled INCENP 160–192 were titrated into $^{13}$C/$^{15}$N-labeled HP1α CSD (Fig. 4 A). When in free state, each CSD of its dimer, having an identical amino acid sequence, was indistinguishable in the spectrum (Fig. 4 A, peaks in black). The addition of the INCENP fragment (160–192 aa) induced significant chemical shift changes in the CSD signals in a series of amino acids, and a pair of two signals were identified (Fig. 4 A, peaks in red and blue), indicating that the binding of INCENP caused an asymmetric environment in the CSD dimer. As the consecutive amino acid assignment in these measurements allowed us to differentiate two species of CSDs, we could tell that the enrichment of peaks around Ile113-Phe117 (region-1) and Ala148-Val151 (region-2) are shifted in only one CSD but not in the other (Fig. 4 B).

These regions were mapped at the side of the CSD dimer and seemed to comprise an interface to associate with INCENP (Fig. 4 C).

These results indicate that HP1α CSD dimer takes an asymmetric mode of interaction in associating with INCENP. In addition, we have identified two signals corresponding to intermolecular nuclear Overhasuer effect (NOE) between one subunit of the CSD dimer labeled by $^{13}$C/$^{15}$N and nonlabeled INCENP; amide proton signals of Ile113, Ala114, and Leu150 from one subunit of the CSD dimer showed strong and weak intermolecular NOE signals with methyl groups of INCENP, suggesting that the stable complex of the CSD dimer bound to INCENP (Fig. S3, E and F). This INCENP/HP1 seems to comprise a group having atypical binding mode because an α-helix is not predicted to assemble next to the PVI motif in many other HP1 CSD-binding proteins (Fig. S3 H; Kang et al., 2011; Kiyomitsu et al., 2010).

### The SSH domain of INCENP is essential for HP1 binding, inner centromere localization of HP1, and the CPC function

Based on these results, we carried out the docking calculations between HP1α CSD dimer and INCENP (160–192 aa) and modeled this HP1-binding domain of INCENP, called the SSH domain (after the Structure composed of β-Strand and α-Helix) (Fig. 5 A). To address the relevance of the α-helix more specifically, we searched for amino acids whose alanine substitution disrupts HP1 binding within the sequence involved in α-helix folding (Fig. 3 H). We found that INCENP bearing point mutation at glutamate 180 (E180) lost its ability to bind to HP1α, to an extent similar to the 179–191 truncated and PVI_3A mutations (Fig. 5 B and Fig. S4 B). Remarkably, the HP1α enrichment at inner centromeres was significantly decreased in mitotic cells expressing the E180A mutant, likewise in those expressing the PVI_3A mutant (17% and 13%, respectively) (Fig. 5 C). These reduced levels of HP1α enrichment were equivalent to that of the INCENP mutant having both PVI_3A and E180A (Fig. S4 C), indicating that the functional SSH domain requires both PVI motif (β-strand) and the α-helix, and that HP1α localizing to the inner centromeres relies on this "bipartite" binding domain of INCENP in mitosis.

The relevance of HP1 binding on mitotic fidelity can be better examined in chromosomally stable, non-transformed cell lines such as RPE1 cells than in cancer cell lines (Abe et al., 2016). So, we first addressed in RPE1 cells to which extent inner centromeric enrichment of HP1 depends on the SSH domain of INCENP and found that its dependency in RPE1 cells was comparable with that in HeLa cells (Fig. S4, D and F). We previously reported that HP1 is required for the CPC to fully convey the kinase activity of Aurora B, and the requirement of this function for

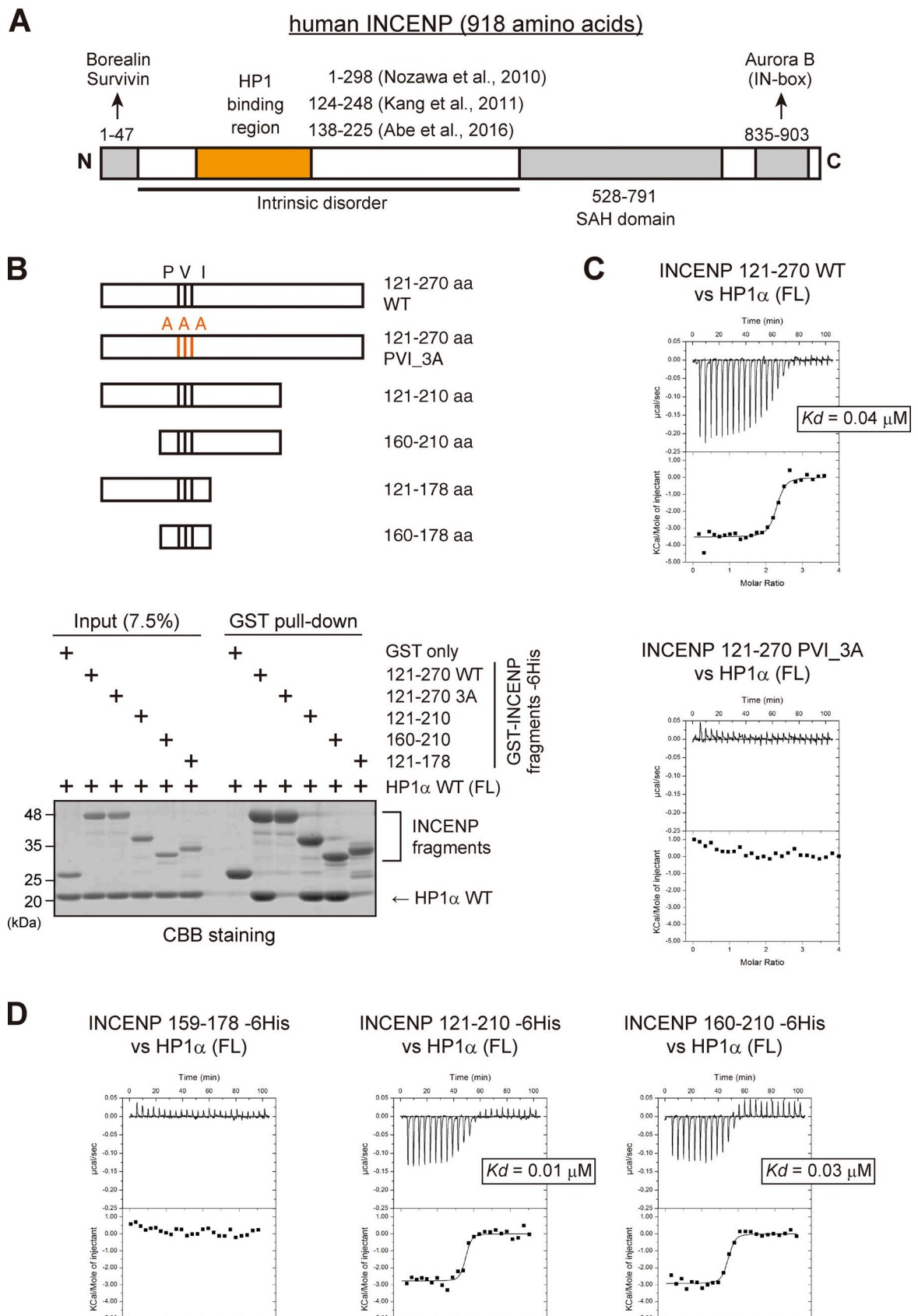

Figure 2. **The sequence next to the PVI motif in INCENP supports the interaction with HP1 *in vitro*. (A)** Schematic representation of human INCENP with known functional regions, including the N-terminal domain (1–47 aa) that binds to Borealin and Survivin, central long intrinsically disordered region, single

alpha helix (SAH) domain (528–791 aa), and the C-terminal Aurora B binding domain called IN-box (835–903 aa). The region supporting the HP1 interaction has been inferred in several studies, as indicated. **(B)** *In vitro* pull-down assay using indicated recombinant proteins. A series of GST-tagged INCENP mutants tested to pull down HP1α (full-length) and representative CBB stained gel is shown. **(C)** Isothermal titration calorimetry (ITC) measurements of the binding between INCENP fragments (121–270, wild type or PVI_3A mutant) and HP1α dimer (full-length, FL). Note that titration with the wild-type INCENP fragment 121–270 revealed an exothermic enthalpy-driven mode of reactions that finally converged, indicating that the interaction occurred in a specific manner. **(D)** ITC measurements between the indicated INCENP fragments and the full-length of HP1α dimer. Source data are available for this figure: SourceData F2.

chromosome segregation is better manifested in chromosomally stable cell lines (Abe et al., 2016). Thus, we analyzed the proficiency of chromosome segregation using RPE1 cells.

An elaborate microscopic analysis of chromosome movements has identified a type of lagging chromosome that emerges transiently and eventually coalesces into the bulk of separating sister chromatids and does not form a micronucleus (Sen et al., 2021). These so-called lazy chromosomes are often found at varying degrees (~20%) even in RPE1, which might reflect missegregation potential and would be a sensitive readout for the Aurora B activity. In these criteria, ~70% of cells underwent error-free segregation, and the vast majority of INCENP-depleted cells revealed serious segregation errors as expected (Fig. 5 D and Fig. S4 G). An exogenous expression of wild-type INCENP (WT) largely rescued these situations, whereas the cells replaced by PVI_3A or E180A mutant could only partially rescue: the ratio of cells with low-lazy chromosomes in these mutants did not grossly change compared with WT, but the incidence of missegregation and high-lazy chromosomes were two times higher than the WT replacement (Fig. 5 D). These data suggest that the lack of HP1 binding to INCENP results in an apparent reduction of Aurora B activity, such that cells underwent mitosis with major KT-MT attachment errors. To verify this idea, we examined the phosphorylation of Hec1 in INCENP E180A cells as a readout for Aurora B activity on kinetochore substrates. As expected, the phosphorylation levels of Hec1 were significantly reduced by ~20%, an extent similar to that in the PVI_3A mutant expressing cells (Fig. 5 E).

## The INCENP SSH domain ensures a stronger interaction with HP1 than a canonical PVI motif

These results demonstrate that the SSH domain of INCENP is required for stable HP1 binding and proficient Aurora B's error-correcting function, thus ensuring mitotic fidelity. As an orthogonal approach to address the significance of the SSH domain, we measured the affinity between HP1 and INCENP (SSH domain) or Sgo1 (canonical PVI motif) by the ITC assay using purified INCENP fragment (residues 121–210) and Sgo1 fragment containing the PVI motif with a comparable length (residues 405–494) (Fig. 6 A and Fig. S5 A). Notably, the heat release from INCENP/HP1α interaction was more than fourfold higher than that from Sgo1/HP1α interaction, and the resulting dissociation constant ($K_d$) of the former was ~60 nM, which is considerably smaller than the latter (Fig. 6 A and Fig. S2 F). These results raise an interesting possibility that an overwhelming affinity of INCENP to HP1 supported by the SSH domain ensures INCENP is the major receptor of HP1 at mitotic inner centromeres (Fig. 1 D). This might also explain the predominant binding of INCENP/HP1 in both the G2 phase and M phase (Fig. 1 E).

To test these possibilities, we expressed INCENP that can ectopically localize at the exterior of centromeres by fusing it to the DNA binding domain of CENP-B (CENP-B box, or CB) (Pluta et al., 1992), and asked if this ectopically expressed CB-INCENP can recruit HP1α *in situ* (Fig. 6 B). To be able to directly assess the effect, we deleted both the N-terminus and C-terminus domains of INCENP (INCENP*) because the former facilitates centromere localization and the latter binds to Aurora B that possibly affects HP1 binding. We found this ectopically expressed CB-INCENP* strongly recruited endogenous HP1α in both interphase and M phase (Fig. 6, C and D). In interphase, expression of CB-INCENP* having SSH domain induced a marked recruitment of HP1α, whereas the intensity of CB-INCENP* Sgo1-PVI_WT was at most half of that of INCENP* WT (Fig. 6 C). Whereas in the M phase, the inner centromere–enriched HP1 normally reveals a single peak, which was now found in two split foci by the expression of CB-INCENP* WT, this effect was highly penetrant unless the SSH domain was mutated (E180A). The Sgo1's PVI motif-substituted version (INCENP* Sgo1-PVI) also lost this effect significantly (Fig. 6 D).

As a counter experiment, we examined if ectopic expression of Sgo1 can also recruit HP1α. To allow for a straightforward interpretation, we additionally mutated Sgo1 at glutamine 61 (Sgo1*) so as not to recruit the phosphatase PP2A-B56 that would compete with the activity of Aurora B (Fig. 6 B; and Fig. S5, B and C; Meppelink et al., 2015). In interphase, expression of CB-Sgo1* having SSH domain induced a marked recruitment of HP1α, whose intensity was ~2.7-fold higher than that induced by CB-Sgo1* WT (Fig. 6 E). Sgo1 failed to recruit HP1α also in mitosis unless the SSH domain derived from INCENP (residues 160–192) was transplanted to and replaced with the PVI motif (residues 447–458) (Fig. 6 F).

These results support the notion that the SSH domain is necessary and sufficient to attain a stable HP1 binding, and the affinity seems to be stronger than the canonical PVI motif. As SSH-mediated recruitment of HP1 was seen in both interphase and mitosis (Fig. 6, C–F), it was conceivable that the SSH domain is functional as long as INCENP is expressed. Consistent with this idea, the E180A mutation disturbed the binding of endogenous HP1 with INCENP in both the G2 phase and M phase (Fig. S5 D).

Finally, to directly address the significance of INCENP having an SSH domain, we asked, what if INCENP had a canonical PVI motif instead of the SSH domain? To address this, we generated RPE1 cell lines that express INCENP with a PVI motif in place of the SSH domain. In the immunoprecipitation assay, we verified that INCENP having a substitution of its SSH domain with Sgo1's PVI motif (INCENP Sgo1-PVI) decreased the stability of HP1 binding, not completely but significantly (Fig. S5 E). The

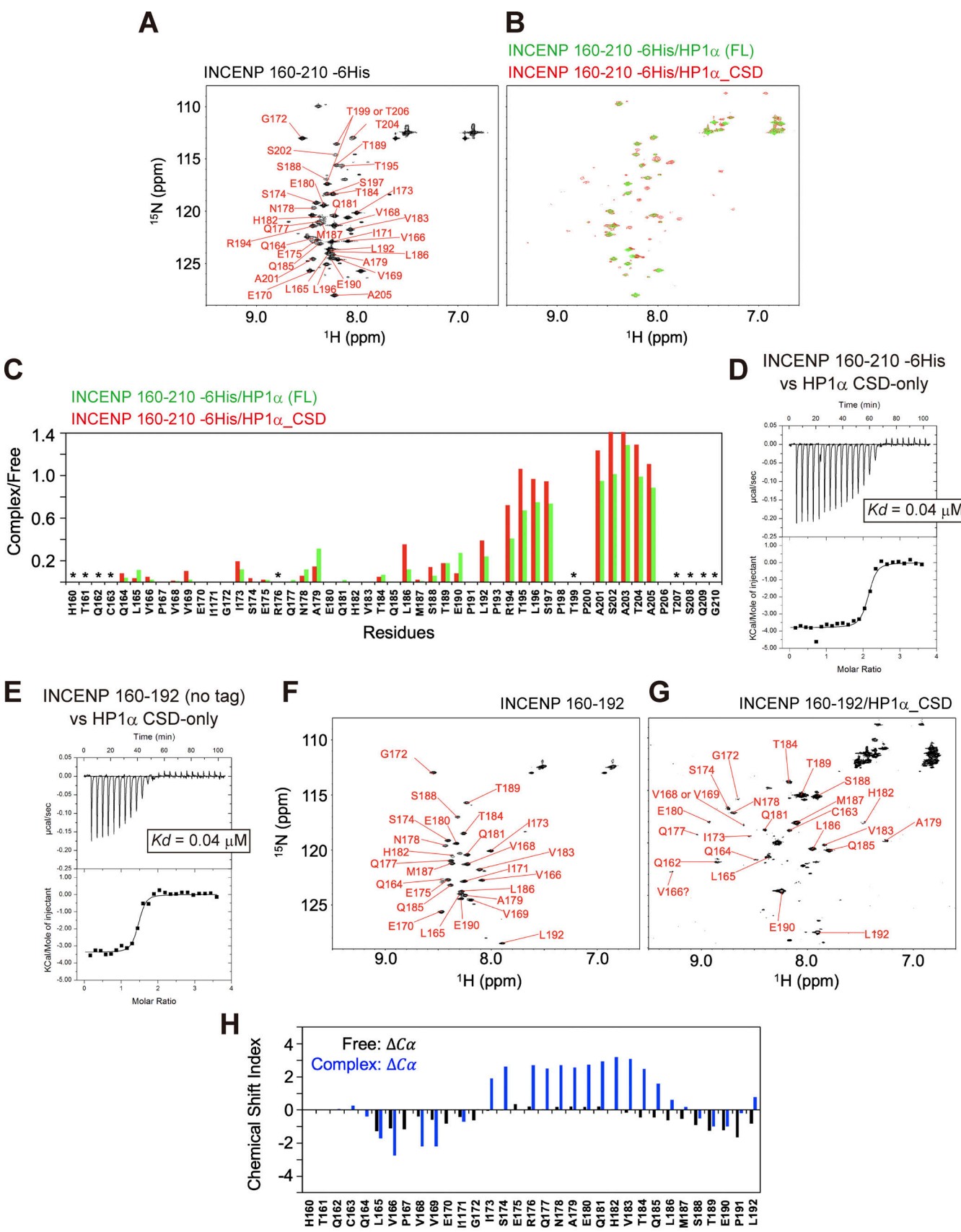

Figure 3. **The disordered sequence next to the PVI motif folds into an α-helix upon HP1 binding. (A)** $^1$H-$^{15}$N HSQC NMR spectrum and amino acid assignments of the INCENP fragment spanning amino acids 160–210. **(B)** Superimposed of $^1$H-$^{15}$N HSQC spectra of INCENP (160–210) in complex with full-

length HP1α dimer (green) and with CSD dimer of HP1α (red). **(C)** NMR measurements of 100 µM INCENP (160–210) in the complex either with 150 µM full-length HP1α dimer (green) or with 150 µM CSD dimer of HP1α (red). Signal intensities relative to those of INCENP alone are shown for each assigned amino acid. Asterisks represent unassigned residues. Both full-length and CSD form a dimer, and the CSD dimer was found to bind to INCENP, with a similar binding affinity as the full-length dimer in ITC measurements (Fig. 2 D and Fig. 3 D). **(D)** ITC measurement of the binding between INCENP (160–210) -6His and HP1α CSD dimer. **(E)** ITC measurement of the binding between the untagged version of INCENP (160–192) and HP1α CSD dimer. **(F)** ¹H-¹⁵N HSQC spectrum and assignments of 100 µM INCENP (160–192). **(G)** ¹H-¹⁵N HSQC spectrum and assignments of 336 µM INCENP (160–192) in the complex with 331 µM CSD dimer of HP1α. **(H)** Chemical shift indices (CSI) of INCENP (160–192) alone calculated from ¹³C$_α$ (black) and INCENP (160–192) in the complex with CSD dimer of HP1α calculated from ¹³C$_α$ (blue). The secondary structure can be predicted by CSI: CSI > 2 indicates α-helix and CSI < −2 indicates β-strand conformations.

chromosome segregation fidelity of the cells expressing this INCENP Sgo1-PVI was perturbed; the combined rate of mis-segregation and high-lazy chromosomes was almost two times higher than cells expressing wild-type INCENP (Fig. 6 G). Accordingly, immunofluorescence microscopy of Hec1 phosphorylation indicated an impaired Aurora B activity in both INCENP Sgo1-PVI and E180A mutants (Fig. 6 H). Of note, incapacitating the PVI motif additionally in INCENP Sgo1-PVI (PVI_3A) did not further affect Aurora B activity and segregation errors, corroborating the significance of the α-helical segment of the SSH domain.

## Discussion

In this study, we identified an HP1 binding interface termed SSH domain in INCENP which confers specific and predominant association among the other HP1 binding proteins. Beyond the prediction with AlphaFold2, which predicted the α-helix formation in the SSH domain (Fig. S3 G), NMR analysis importantly indicates that it is conditionally assembled from the disordered sequence upon HP1 binding (Fig. 3 H and Fig. S3 D). Moreover, the exact amino acids linking the α-helical segment and HP1α CSD can be identified through the NOE measurements (Fig. S3, E and F). Thus, the predicted structure is useful yet requires experimental verifications.

Given that the PVI motif, like in many other HP-interacting proteins, plays a major role in INCENP binding to HP1, it was unexpected to find that fragments containing the PVI motif do not bind to HP1 unless they have an α-helical segment (Fig. 2). To what extent do PVI and α-helical segments individually support the binding? The ITC measurements provide insights into this question: Comparison of INCENP 121–270 fragment wild-type (dissociation constant $Kd$ = 0.042 µM) with its 3A counterpart ($Kd$: N/A) revealed not only the requirement of PVI motif but also that the α-helical segment is insufficient for binding. Conversely, comparison of INCENP 121–210 wild-type fragment ($Kd$ = 0.06 µM) with the fragment bearing E180A mutant ($Kd$ = 0.81 µM) revealed not only a higher affinity of PVI motif but also a supporting role of the α-helical segment in establishing a stable interaction (Fig. 6 A; and Fig. S5 A). Thus, the INCENP PVI motif dose not bind to HP1 on its own; however, together with the downstream α-helix, INCENP creates a high affinity for HP1, underscoring the significance of the bipartite binding interface.

On what structural basis can neither the PVI motif nor the α-helical segment of INCENP alone bind to HP1? A study of the CAF-1/HP1 complex indicated that hydrophobic residues flanking both sides of the PVI motif are also recognized by HP1, additionally to PVI *per se* (Thiru et al., 2004). These hydrophobic residues are in fact present at the corresponding position in representative HP1 interactors but those residues are less conserved in INCENP (Fig. S2 B). In addition, INCENP seems to lack an electrostatic bond between a PVI-flanking residue and the CSD pointed out in Sgo1/HP1 (Kang et al., 2011). Furthermore, based on our calculation of chemical shift indices (Fig. 3 H), the β-strand segment spanning the PVI sequence could be shorter than that formed in Sgo1 and CAF-1 (Kang et al., 2011; Thiru et al., 2004). These characteristics of INCENP's PVI motif may account for its atypically low affinity to HP1. The chemical shift indices also indicated that the α-helix structure is assembled in an inducible manner upon interacting with HP1 (Fig. 3 H and Fig. S3 D). Together with the finding that the α-helical region did not support the HP1 binding of the PVI_3A mutant (Fig. 2 C and Fig. 5 B), the α-helical region might not form its secondary structure by itself.

It is noteworthy that the INCENP/HP1 binding is reminiscent of the interaction of Mit1, a subunit of the remodeler SHREC complex, with HP1 (Swi6) ortholog Chp2 in fission yeast. The binding interface in Mit1 consists of two distinct domains, the PVI motif and subsequent globular structure called chromodomain-like domain, which is required for the stable interaction with Chp2 (Leopold et al., 2019). The HP1-binding interface in INCENP did not contain such a globular domain but instead, an α-helical module was assembled from a disordered sequence conditionally upon binding to HP1 (Figs. 3 and 4). Such conditional folding of the disordered sequence has been found in several proteins, including the pKID domain of CREB, the TAD domain of p53, the SRM domain of EZH2, and the AIL1 domain of separase (Borcherds et al., 2014; Poepsel et al., 2018; Radhakrishnan et al., 1997; Yu et al., 2021). Importantly, these foldings play a critical role in ensuring specific interaction for each protein, as SSH domain does for INCENP, implying the general use of the conditional interaction module for proteins with disordered sequences.

Thus, there exist at least three different modes for the interaction between HP1 and its binding proteins, namely using the canonical PVI motif (e.g., Sgo1 and HP1α), conditional bipartite interface (INCENP and HP1α), and two distinct domains (Mit1 and Chp2). A comparison of their dissociation constant suggests different modes of interaction; $Kd$ for Sgo1 and HP1α ranges from 180 to 4,000 nM (Fig. 6 A; Kang et al., 2011), INCENP and HP1α is ~60 nM (Fig. 6 A), and Mit1 and Chp2 is 2.6 nM (Leopold et al., 2019). These measurements also indicate that bipartite binding interfaces reinforce the binding affinity additionally to its level achieved through the PVI motif. We could reason that these differences provide proper and

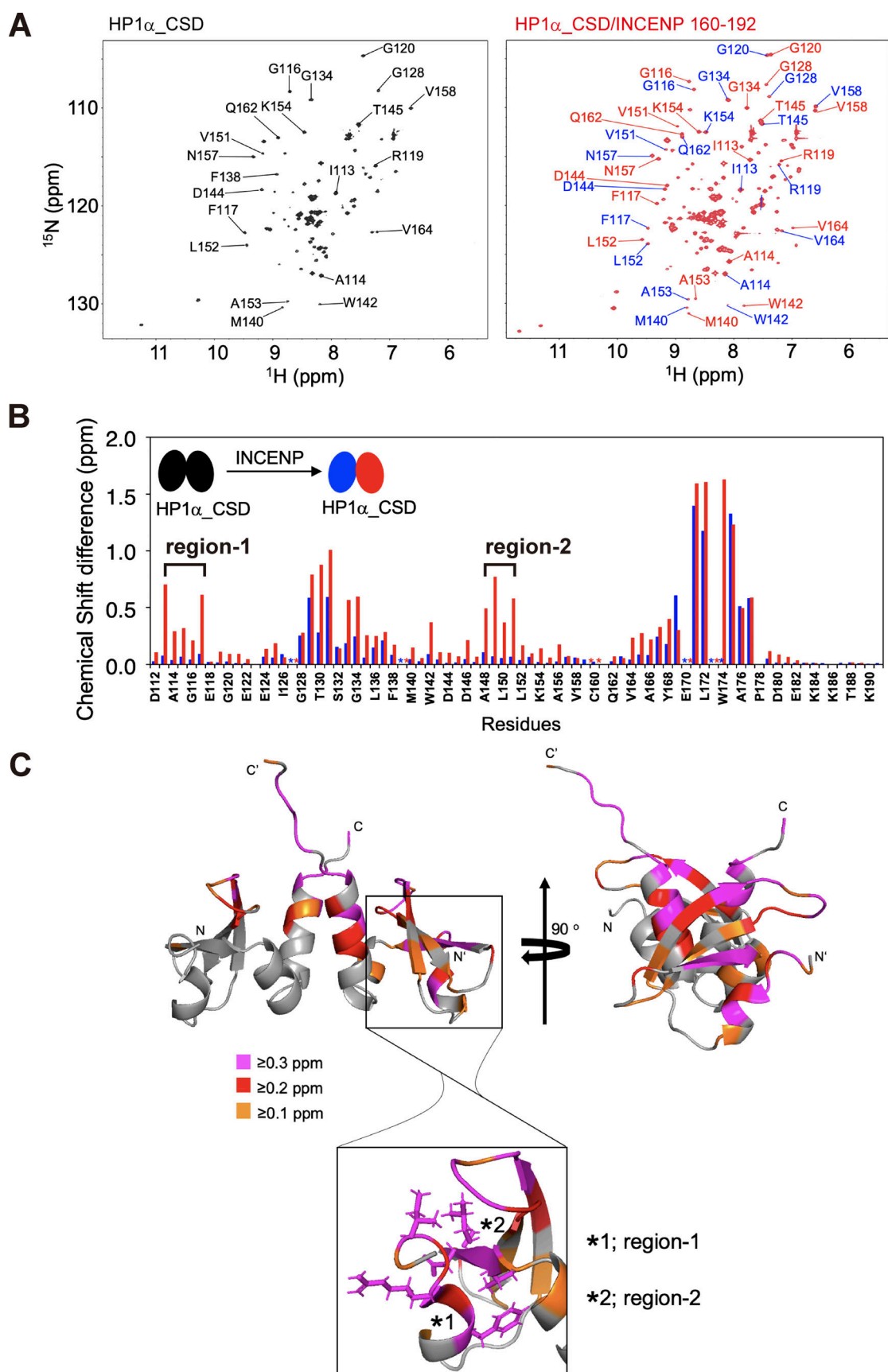

Figure 4. **A lateral surface of CSD dimer mediates the binding to INCENP. (A)** The ¹H-¹⁵N HSQC spectra of HP1α CSD dimer alone (left panel) and that of 1.3 mM CSD dimer bound to 1.6 mM INCENP (160–192) (right panel, red) are shown. During the titration, we found the signals from the CSD dimer disappeared

and new signals from the INCENP-bound CSD emerged. Some examples of residues separated into two peaks are listed on the spectrum (right panel). **(B)** Chemical shift differences between CSD dimer alone and INCENP-bound CSD dimer. The consecutive amino acid assignment allows distinguishing each CSD molecule comprising the dimer, as differentiated in blue and red bars. Note that the binding of INCENP caused resonance scattering on only one CSD in ranges denoted by region-1 and -2. **(C)** Chemical shift differences determined in B are demonstrated as a color-coded secondary structure of the CSD dimer. The amino- and carboxyl-termini of the two CSD molecules are denoted by N/C and N'/C', respectively.

appropriate levels of stability for their interactions. For example, we have learned that INCENP's higher affinity to HP1 than the affinity achieved by PVI motif alone is required to fulfill the function of the CPC (Fig. 6). As excess phosphorylation of kinetochore substrates of Aurora B lead to a defect in the process of microtubule attachments to kinetochores (Liu et al., 2009), we can speculate that too stable binding of INCENP with HP1 might rather be deleterious. An appropriate strength of HP1 interaction is required for the proper function of the CPC, and the SSH domain conceivably allows achieving such a level of interaction.

To what extent is the HP1–CPC interaction conserved through evolution, and how does its binding mode differ from other HP1 interactors? Based on the secondary structure predictions, the SSH-like domain can be found in vertebrate INCENP homologs (Fig. S3 I). In non-vertebrates, the PVI motif becomes less clear, and the SSH domain-like structure is no longer evident (data not shown). Despite these predictions, whether the HP1-mediated regulation of the CPC operates in other organisms remains unclear. Capitalizing on AlphaFold2 predictions, we could assume that HP1 interactors can be separated into two groups (Fig. S3 H). One group adopts a bipartite binding interface similar to the SSH domain (e.g., hINCENP and hScc2) and the other uses the canonical binding mode, relying solely on the PVI motif (e.g., hSgo1, hMis14, hCAF1-A, and hTIF1β). These predictions allow us to propose that the bipartite interface is an unconventional binding mode among the HP1 interactors.

The SSH domain of INCENP mediates the binding to HP1 continuously from the G2 phase to mitosis. The finding that the SSH domain alone could sufficiently recruit HP1 in both G2 and M phases indicates that it does not require mitotic phosphorylation to fulfill its role (Fig. 6 and Fig. S5 D). In this vein, in the G2 phase, the interaction seems to take place in the absence of Aurora B-mediated phosphorylation of serine 92 in the hinge domain of HP1α (Abe et al., 2016), i.e., before Aurora B attains its kinase activity (Fig. S5 D). This implies that the interaction occurs irrespectively of Aurora B activity, which would allow CPC to be recruited to HP1 in interphase that had been associated with trimethylated lysine 9 of histone H3 (H3K9me3) (Bannister et al., 2001; Jacobs et al., 2001; Lachner et al., 2001). Upon mitotic entry, the elevated activity of Aurora B phosphorylates H3 serine 10 adjacent to lysine 9 (H3K9me3S10ph) and facilitates the dissociation of HP1 from chromosome arms (Fischle et al., 2005; Hirota et al., 2005), and HP1-bound CPC in turn becomes recruited to phospho-marked H3 (H3T3ph) at centromeres. HP1-associated proteins with the canonical PVI motif, such as Sgo1 and Nls1/Mis14, are known to occur primarily with HP1 in interphase (Kang et al., 2011; Kiyomitsu et al., 2010), whereas SSH domain becomes essential in M phase to maintain the robust

INCENP/HP1 interaction and the enrichment of CPC at inner centromeres (Fig. 6).

It seems however that the mode of interaction between HP1 and CPC changes drastically from G2 to M phase in several ways. Of note, we previously found that HP1α carrying non-phosphorylatable mutant at serine 92 (S92A) reveals reduced stability for binding to INCENP in the M phase. Thus, unlike the initial interaction in the G2 phase, the enrichment of HP1-bound CPC at inner centromeres seems to involve mitotic phosphorylation of HP1's hinge domain. Given that this phosphorylation at serine 92 is mediated by Aurora B, and it in turn stabilizes the association with the CPC, it is plausible that an indirect positive feedback between HP1 and CPC contributes to maintaining CPC and the Aurora B activity at inner centromeres. Indeed, the hinge domain of HP1 is known to have the ability to associate with RNAs in vitro (Muchardt et al., 2002), and an attractive hypothesis would be that phosphorylation of the hinge domain facilitates the interaction with RNA molecules. These possibilities are consistent with the findings that RNA has been implicated in the function of Aurora B at inner centromeres (Blower, 2016; Chan and Wong, 2012; Ferri et al., 2009).

## Materials and methods
### Cell culture, cell lines, lentivirus, and RNAi
Cells (HeLa-Kyoto and hTRET-RPE1) were cultured in DMEM supplemented with 10% FBS, 0.2 mM L-glutamine, 100 U/ml penicillin, and 100 μg/ml streptomycin at 37°C in a 5% $CO_2$ environment. EGFP-CENP-A expressing HeLa cell lines were generated in previous studies (Gerlich et al., 2006; Kunitoku et al., 2003). To generate HeLa cell lines that stably express INCENP -6Myc series, HeLa cells were transfected with pIRESpuro3-human INCENP (encoding INCENP siRNA-resistant synonymous substitutions (lower case); 5'-CTCCGTCGAAAA ATTAGCCACCGTG-3') of full-length wild-type (WT), PVI_3A (P167A/V169A/I171A), Δ179-191, E180A, or PVI_3A+E180A mutant by using Lipofectamine 3000 transfection kit (Invitrogen). Stably expressing cell clones were selected in a growth medium containing 0.25 μg/ml puromycin and verified by immunofluorescence microscopy and immunoblot for the expression of the tagged transgene. RPE1 cell lines that stably express INCENP -6Myc WT or PVI_3A mutant were generated by lentivirus infection in our previous study (Abe et al., 2016). Likewise, INCENP -6Myc E180A, PVI_3A+E180A, Sgo1-PVI_WT, and Sgo1-PVI_3A mutant RPE1 cell lines were also generated by lentivirus infection in this study. Lentiviruses were generated by transfecting modified pLenti6.3/V5-DEST vectors (Invitrogen) encoding INCENP -6Myc series together with the lentiviral packaging vectors (ViraPower Lentiviral Packaging Mix, Invitrogen) to HEK293T cells. Lentiviral culture medium

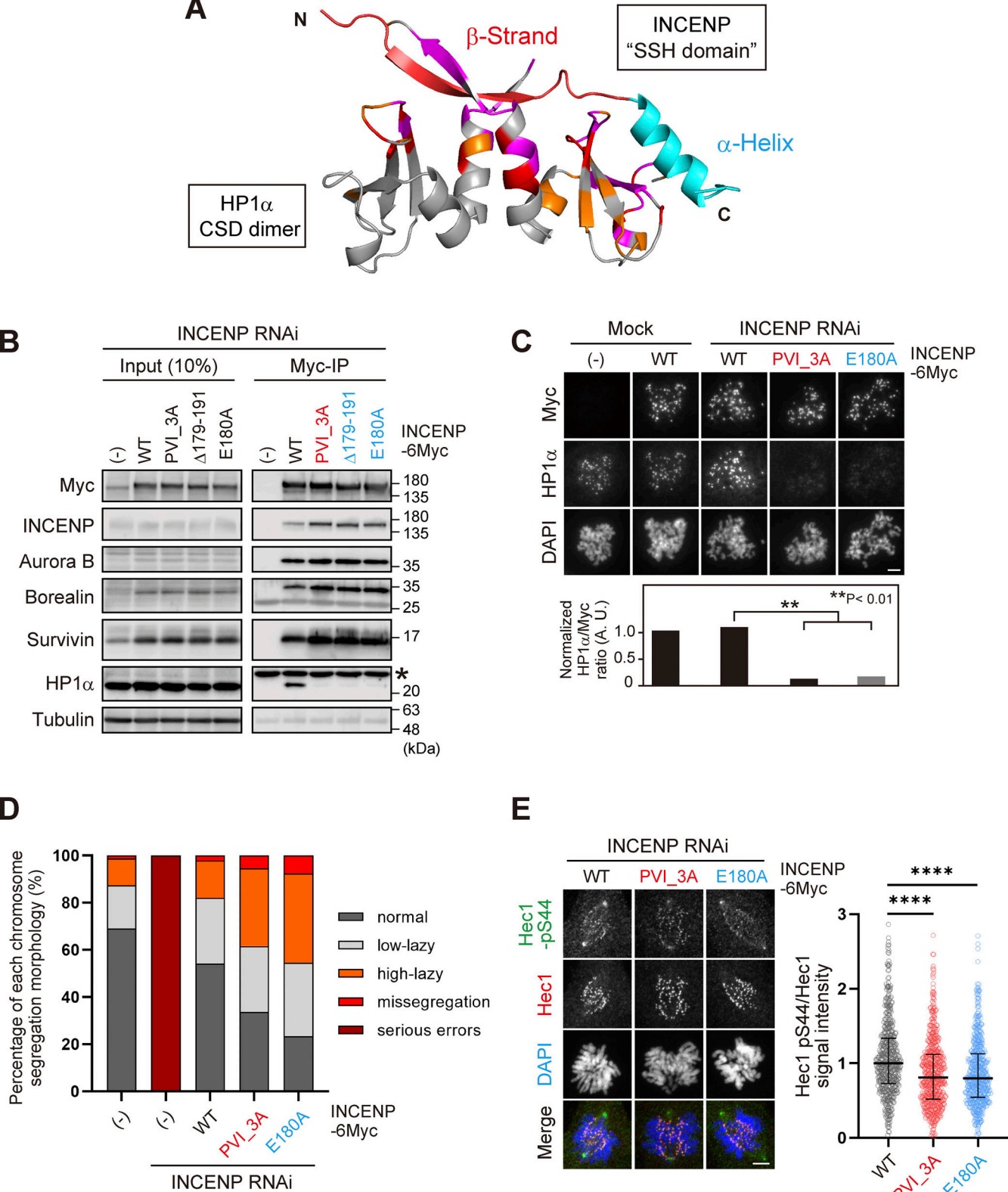

Figure 5. **The SSH domain is essential for HP1 binding, inner centromere recruitment of HP1, and CPC function. (A)** A docking model between HP1α_CSD (111–179) dimer and INCENP (160–192) using the HADDOCK (de Vries et al., 2010; Wassenaar et al., 2012). Chemical shift differences determined in Fig. 4 B are demonstrated as a color-coded secondary structure of the CSD dimer. The β-strand including the PVI motif and α-helix constituting the SSH domain are colored in red and light blue, respectively. **(B)** Mitotic extract prepared from HeLa cells expressing wild type (WT) or various mutants of INCENP -6Myc in place of endogenous INCENP were subjected to immunoprecipitation with anti-myc antibodies and immunoblotted with indicated antibodies. (–) indicates the parental cells. An asterisk indicates nonspecific proteins (light chain of IgG). Note that three SSH mutants can assemble the whole CPC complex

but largely lack the ability to bind to HP1α. **(C)** Immunofluorescence microscopy of chromosome spreads prepared from HeLa cell lines expressing the indicated version of INCENP -6Myc. Cells were transfected with an siRNA to INCENP for 48 h and treated with 100 ng/ml nocodazole for 2 h. Fixed cells were stained with antibodies to myc and HP1α. Similar results were obtained in RPE1 cell lines (shown in Fig. S4 D). Scale bar, 5 µm. **(D)** RPE1 cell lines stably expressing indicated versions of INCENP were depleted of endogenous INCENP and examined for chromosome segregation in anaphase, following the release from 0.2 µM palbociclib treatment. Anaphase cells were classified based on their morphologies exemplified in Fig. S4 G. Each histogram shows the average of two independent experiments, and a total of at least 130 cells were assessed. **(E)** Immunofluorescence microscopy of Hec1-pS44 in INCENP WT, PVI_3A, or E180A expressing RPE1 cells, replacing the endogenous INCENP. Relative fluorescence intensities of Hec1-pS44 being normalized to Hec1 are shown in the beeswarm plots. For all cell lines, more than 550 kinetochores were quantified from at least 14 cells. The black bars represent the median with the interquartile range. P values were calculated with the Mann–Whitney test (****P < 0.0001). Scale bar, 5 µm. Source data are available for this figure: SourceData F5.

---

was collected 72 h after transfection and added to RPE1 cell lines. Cell clones were selected in a growth medium containing 20 µg/ml blasticidin and were verified by immunofluorescence microscopy and immunoblot for the expression of the tagged transgene. The transient expression experiments were conducted as follows: In Fig. S4 B, HeLa cells were transfected with pIRESpuro3-human INCENP -6Myc of full-length WT, PVI_3A, E180A, Q181A, or H182A mutant; in Fig. 6, D and F, HeLa cells were transfected with pcDNA5/TO-Myc-CB-fused INCENP or Sgo1 constructs (detailed in Plasmids); in Fig. S5, B and C, EGFP- CENP-A expressing HeLa cells were also transfected with the above indicated pcDNA5/TO-Myc-CB-fused constructs. All transient expressing cells were collected or fixed after 48 h expression. For RNAi depletion of INCENP, Sgo1, and HP1α, cells were treated with specific siRNAs using RNAiMax (Invitrogen) for 24 h (INCENP and Sgo1) or 48 h (HP1α) (detailed in RNA interference).

### Antibodies

Polyclonal rabbit antibodies to human INCENP were raised against two synthetic peptides (C+RRKSRSSQLSSRRL, 88–101 amino acids; C+ARVPSSLAYSLKKH, 905–918 amino acids). Anti-INCENP antibodies were affinity-purified using the mixed antigen. Polyclonal rabbit antibodies to human HP1α-phospho-Ser92 were generated in our previous study (Abe et al., 2016). Polyclonal rabbit antibodies to human CENP-A-phospho-Ser7 were generated in our previous study (Kunitoku et al., 2003). Human CREST sera were used as described (Abe et al., 2016). Polyclonal rabbit antibodies to human Sgo1 and monoclonal rat antibodies to human HP1α were kindly provided by Ana Losada (Spanish National Cancer Research Centre [CNIO], Madrid, Spain). Polyclonal rabbit antibodies to human Hec1-phospho-Ser44 were kindly provided by Jennifer G. DeLuca (Colorado State University, USA). We used rabbit polyclonal antibodies against INCENP (#2807S; Cell Signaling Technology), Sgo2 (#A301-262A; BETHYL), Borealin (#NBP1-77330; Novus Biologicals), Phospho-cdc2 (Tyr15) (#9111S; Cell Signaling Technology), Myc-tag (#562; MBL), and rabbit monoclonal antibody against Survivin (clone 71G4B7, #2808S; Cell Signaling Technology). We used guinea pig polyclonal antibody against CENP-C (#PD030; MBL), rat monoclonal antibody against Myc-tag (clone 9E10, #ab206486; Abcam), and mouse monoclonal antibodies against Sgo1 (clone 3C11, #ab58023; Abcam), Aurora B (AIM-1, clone 6; #611083; BD Biosciences), HP1α (clone 15.19s2, #05-689; Millipore), HP1β (clone 1MOD-1A9, #MAB3448; Millipore), HP1γ (clone 14D3.1, #MABE656; Millipore), Cdc27 (clone 35, #610454; BD Biosciences), α-Tubulin (clone B-5-1-2, #T6074, Millipore), Myc-tag (clone 4A6, #05-724;

Millipore), and HEC1 (clone 9G3, #ab3613; Abcam). We used Myc-tag mAb-Magnetic Beads (clone PL14, #M047-11; MBL) and rabbit control IgG (#I5006-10MG; Sigma-Aldrich).

### Plasmids

Plasmids encoding wild-type or PVI_3A INCENP -6Myc (pIRES) were generated as described (Abe et al., 2016). To generate pIRESpuro3- or pLenti6.3/V5-DEST-plasmids encoding INCENP Δ179-191, E180A, PVI_3A+E180A, Sgo1-PVI_WT, and Sgo1-PVI_3A, we used an In-Fusion HD Cloning kit (TaKaRa) and performed cloning using the above plasmids as a template (Figs. 5, 6, S4, and S5). In Fig. 6, C–F and Fig. S5, B and C, plasmids encoding Myc-CB-fusion proteins were generated as follows: INCENP DoubleΔ (INCENP*) mutants were generated by deleting the N-terminal Survivin- and Borealin-binding domain and the Aurora B-binding IN-box. These domains may promote the localization of INCENP to inner centromeres and thus were removed to allow straightforward interpretations of the experiment: the former mediates the inner centromere localization of INCENP and the latter contributes to HP1α localization due to the positive feedback circuit between Aurora B and HP1α (Abe et al., 2016). The Sgo1 mutants substituted with the SSH domain were generated by replacing Sgo1 447-458 (12 aa) with the INCENP-SSH domain (33 aa).

### Cell synchronization

To obtain mitotic HeLa cells, cells were cultured with 7.5 µM STLC for 16 h, shaken off, and collected from culture dishes unless otherwise stated. For RPE1 cells, cells were cultured with 0.2 µM palbociclib for 24 h, and after 3 h from the release, mitotic cells were enriched by treating cells with 7.5 mM STLC for 15 h. To obtain G2 phase–enriched HeLa cells, double thymidine-synchronized cells were treated with 10 µM RO-3306. For immunofluorescence microscopy, 24 h after transfection, cells were replated onto cover glasses, treated either with 2 mM thymidine for 20 h, and fixed at 9.5 h after the release (Fig. 6, C–F and Fig. S5 B), or with 100 ng/ml nocodazole for 14 h before the fixation (Fig. S5 C). For mitotic enrichment of RPE1 cells (Fig. 5, D and E; and Fig. 6, G and H; and Fig. S4 G), cells were cultured with 0.2 µM palbociclib for 24 h, with or without RNAi, and cultured for another 15 h before fixation.

### RNA interference

Transfection of siRNAs was performed with 20–100 nM siRNAs using Lipofectamine RNAi MAX Transfection Reagent (Thermo Fisher Scientific) for 24–48 h. The target sequences of siRNA (Stealth; Invitrogen) for INCENP (5′-CAGUGUAGAGAAGCUGGC

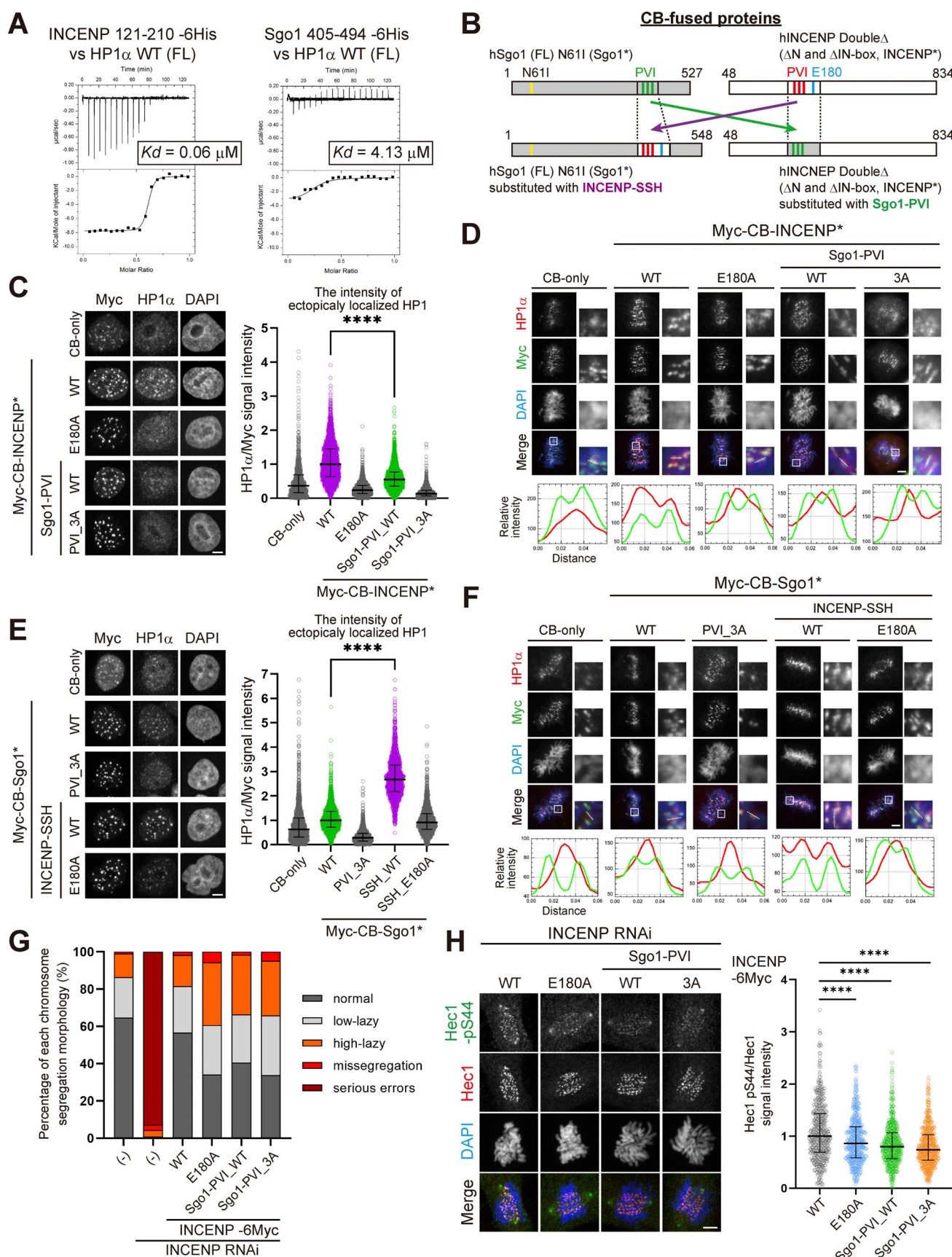

Figure 6. **The SSH domain enables stronger interaction than a canonical PVI motif does. (A)** ITC measurements between recombinant INCENP fragment and HP1α dimer (left) and between Sgo1 fragment and HP1α dimer (right). **(B)** The schematic representation of the CB-fused hybrid mutants of INCENP and

Sgo1. **(C)** Immunofluorescence microscopy of HP1α in HeLa cells transiently expressing Myc-CB-only or indicated mutants of Myc-CB fused INCENP*. Relative fluorescence intensities of HP1α being normalized to myc are shown in the beeswarm plots. For each case, more than 940 foci were quantified from at least 70 cells. The black bars represent the median with interquartile range. P values were calculated with the Mann–Whitney test (****P < 0.0001). Scale bar, 5 µm. **(D)** Immunofluorescence microscopy of chromosome spreads prepared from HeLa cells transiently expressing Myc-CB fused INCENP* mutants. Smaller panels show magnified views of kinetochore pairs of the boxed regions used for line scan measurements. Scale bar, 5 µm. **(E)** Immunofluorescence microscopy of HP1α in HeLa cells transiently expressing either Myc-CB-only or indicated mutants of Myc-CB fused Sgo1*. Relative fluorescence intensities of HP1α were quantified and summarized as in C. For each case, more than 1,680 foci were quantified from at least 60 cells. The black bars represent the median with the interquartile range. P values were calculated with Mann–Whitney test (****P < 0.0001). Scale bar, 5 µm. **(F)** Immunofluorescence microscopy of chromosome spreads prepared from HeLa cells transiently expressing Myc-CB fused Sgo1* mutants. Smaller panels show magnified views of kinetochore pairs of the boxed regions used for line scan measurements. Scale bar, 5 µm. **(G)** RPE1 cell lines stably expressing indicated versions of full-length INCENP were depleted of endogenous INCENP and examined for chromosome segregation in anaphase, following the release from 0.2 µM palbociclib treatment. (−) indicates the parental cells. Chromosome missegregation in anaphase was analyzed as in Fig. 5 D. Each histogram shows the average of two independent experiments, and a total of at least 170 cells were assessed. **(H)** Immunofluorescence microscopy of Hec1-pS44 in the indicated version of full-length INCENP expressing RPE1 cells, in place of endogenous INCENP. Relative fluorescence intensities of Hec1-pS44 being normalized to Hec1, are shown in the beeswarm plots. For each case, more than 550 kinetochores were quantified from at least 14 cells. The black bars represent median with interquartile range. P values were calculated with Mann–Whitney test (****P < 0.0001). Scale bar, 5 µm.

UACAGUG-3′), HP1α (5′-UAACAAGAGGAAAUCCAAUUUCUCA-3′), and Sgo1 (5′-CCCAAUAGUGAUGACAGCUCCAGAA-3′) (all of these are against ORF) were previously described (Abe et al., 2016; Nakajima et al., 2007). For controls, the same reaction was set up using H$_2$O instead of siRNA oligos.

## Immunoblotting

Cells were lysed in a lysis buffer (20 mM Tris at pH 7.5, 150 mM NaCl, 20 mM disodium β-glycerophosphate pentahydrate, 5 mM MgCl$_2$, 0.1% NP-40, 5% glycerol, 1 mM DTT, 0.1 µM okadaic acid) supplemented with 1,000 U/ml OmniCleave Endonuclease (Biosearch Tech.) to extract chromatin-bound proteins, a cocktail of protease inhibitors (Complete EDTA-free, Roche), and phosphatase inhibitors (PhosSTOP, Roche) (Fig. 1 E, Fig. S1, G and H, Fig. S4 F, and Fig. S5 E), and incubated for 30 min at 4°C. After removing the insoluble fraction by centrifugation twice at 15,000 rpm for 10 min, supernatants were collected and their total protein concentration was measured by the Bradford method (Protein Assay system, Bio-Rad Laboratories). The same concentration of supernatants was resolved by SDS-PAGE and transferred to a PVDF membrane and stained with Ponceau S [5% acetic acid solution containing 0.1% Ponceau S] for 1 min. After decolorization by shaking in dH$_2$O, the membrane was incubated with primary antibodies diluted in Can Get Signal Immunoreaction Enhancer Solution 1 (TOYOBO). The horseradish peroxidase-linked secondary antibodies (GE Healthcare) were developed by chemiluminescence using luminol and coumaric acid (Sigma-Aldrich). Luminescent signals were detected with Odyssey (LI-COR) and analyzed using Image Studio software (LI-COR).

## Immunoprecipitation

Cell extracts were prepared as described above, except for Fig. 1 E, in which NaCl was used at the concentration of 500 mM. The supernatants were incubated with the following antibody-linked beads for 1 h at 4°C: anti-Myc magnetic beads (MBL) (Fig. 5 B, Fig. S4, A and B, and Fig. S5 D), Dynabeads Protein G (Invitrogen) conjugated with anti-Myc antibody (clone 4A6; Millipore) (Fig. S4 F and Fig. S5 E), or with custom made antibodies to Sgo1 and INCENP, or rabbit control IgG (Sigma-Aldrich) (Fig. 1 E and Fig. S1 H). The beads were washed three times with the lysis

buffer. For immunoblotting of the immunoprecipitation samples, horseradish peroxidase–linked mouse TrueBlot ULTRA or rabbit/rat TrueBlot (Rockland) was used as secondary antibodies to minimize crosses to IgGs.

## Immunofluorescence microscopy

Cells grown on coverslips were fixed either with 2% paraformaldehyde in NaPO$_4$ (pH 7.4) buffer for 10 min (Hec1-pS44 staining in Fig. 5 E and Fig. 6 H) or with 4% paraformaldehyde in NaPO$_4$ buffer for 20 min (assessment of chromosome segregation in Fig. 5 D, Fig. 6 G, and Fig. S4 G). For HP1α staining in Fig. 6, C and E, cells were pre-fixed with 4% paraformaldehyde in NaPO$_4$ buffer for 20 s and pre-extracted sequentially with PBS containing 0.01% Triton X-100 (PBS-T) for 2 min and with 0.2% Triton X-100/PBS for 2 min, followed by 4% paraformaldehyde fixation for 20 min. For verification of expression, cells were fixed with 4% (Fig. S5 B) or 2% (Fig. S5 C) paraformaldehyde. All samples were permeabilized with 0.2% Triton X-100/PBS for 5–10 min. After blocking with 3% BSA in PBS-T for 30 min to 1 h, cells were incubated with the primary antibodies diluted in 3% BSA in PBS-T. The primary antibodies were used in the following dilution: Hec1-phospho-Ser44 (1:3,000; a gift from Jennifer DeLuca) (Fig. 5 E and Fig. 6 H); Hec1 (1:5,000, clone 9G3; Abcam) (Fig. 5 E and Fig. 6 H); Myc (1:500; MBL) (Fig. 6, C and E; and Fig. S5 B); HP1α (1:300, clone 2HP-1H5; Millipore) (Fig. 6, C and E); CENP-C (1:3,000; MBL) (Fig. S4 G); INCENP (1:1,000, P240; Cell Signaling Technology) (Fig. S4 G); Aurora B (1:500; BD Biosciences) (Fig. S5 B); Myc (1:500, clone 9E10; Abcam) (Fig. S5 C); CENP-A-phospho-Ser7 (1:1,000, custom made [Kunitoku et al., 2003]) (Fig. S5 C). The following secondary antibodies were used: goat anti-rabbit IgG Alexa-Fluor-488, -568, and -647; goat anti-mouse IgG Alexa-Fluor-488 and -568; goat anti-rat IgG Alexa-Fluor-568; goat anti-guinea pig IgG Alexa-Fluor-568 (all from Molecular Probes). After counterstaining DNA with 0.1 µg/ml DAPI, cells were mounted with ProLong Gold anti-fade mounting reagent (Invitrogen). Images of Hec1-pS44 staining (Fig. 5 E and Fig. 6 H) were acquired with a Plan-Apochromat 100×/1.46NA Oil objective lens (Zeiss) on LSM880 confocal laser scanning microscope (Zeiss) (acquisition software is Zen 2.1 SP3 FP3, black edition, version 14.0.25.201); HP1α staining (Fig. 6, C and E) and missegregation assay (Fig. 5 D, Fig. 6 G, and Fig. S4 G)

were acquired with a UPLFLN 60×/0.9NA Dry objective lens (Olympus) on Confocal Quantitative Image Cytometer (CQ1; Yokogawa) (acquisition software is CQ1 software, version 1.07.01.01); verification of expression (Fig. S5, B and C) were acquired with a Plan-Apochromat 63×/1.4NA Oil objective lens (Zeiss) on AxioImagerM1 microscope (Zeiss) equipped using the Prime BSI sCMOS camera (Teledyne) (acquisition software is Zen, blue edition, version 2.6). All the images of fixed cells were acquired by projecting multiple z-planes at the interval of 0.4–0.5 µm. Quantifications of fluorescence signals were conducted as described below.

### Chromosome spreads
Cells were treated with 100 ng/ml nocodazole for 2 h (HeLa) or with 150 ng/ml nocodazole for 12 h (RPE1) (Fig. 5 C; and Fig. S4, C and D). Mitotic cells were shaken-off, washed with PBS supplemented with 200 ng/ml nocodazole, and spun onto glass slides (Star frost, New Silane; Muto) at 1,500 rpm for 5 min using Cytospin centrifuge (Shandon). To examine unperturbed mitotic cells in a native condition (Fig. 1, A–C, Fig. 6, D and F; and Fig. S1, B–F), cells synchronized at G1/S phase by 2 mM thymidine block (HeLa) or 0.2 µM palbociclib (RPE1) treatment for 24 h were harvested at 9.5 h (HeLa) or 15 h (RPE1) after the release. Trypsin-detached cells were suspended in PBS, and spun onto cover glasses (18 mm diameter; Marienfeld) at 1,500 rpm for 5 min with Cytospin. Samples were treated with 1% paraformaldehyde in PBS containing 0.2% Triton X-100 for 10 min, with 50 mM $NH_4Cl$ for 30 min, and then with 0.2% Triton X-100 in PBS for 10 min. After blocking with 3% BSA in PBS for 1 h, the samples were incubated with the primary antibodies in 3% BSA in PBS, overnight at 4°C, followed by incubation with secondary antibodies in 3% BSA in PBS for 40 min at room temperature. The primary antibodies were used in the following dilutions: INCENP (1:400, P240; Cell Signaling Technology) (Fig. 1, A and C; and Fig. S1, B, D, and F); Sgo1 (1:400; a gift from Ana Losada) (Fig. 1 B; and Fig. S1, E and F), Sgo1 (1:400; Abcam) (Fig. 1 C); HP1α (1:200; a gift from Ana Losada) (Fig. 1, A and B, Fig. 6, D and F, and Fig. S1, B, C, and F); Myc (1:200; MBL) (Fig. 6, D and F); and CREST sera (1:1,000) (Fig. S1, C–F). Secondary antibodies, DAPI staining, mounting, and image acquisition using AxioImagerM1 microscope (Zeiss) were conducted as described above.

### Recombinant proteins
pGEX6P1 plasmids encoding GST-tagged human INCENP (hINCENP)-6His fragments (121-270 WT, 121-270 PVI_3A, 121-210, 160-210, and 121-178) (Fig. S2 D and Fig. S3 A), GST-tagged Tyr-fusion hINCENP fragments (160-192, 160-210), and hSgo1 fragment (405-494), in which tyrosine residue was inserted between PreScission site and INCENP for determination the concentration (Fig. S3 C and Fig. S5 A), were transformed in *E. coli* BL21 Codon Plus (Stratagene). pCold plasmids encoding 6His-MBP-tagged full-length hHP1α WT (Fig. S2 C) and 6His-tagged hHP1α CSD (Fig. S3 B) were transformed in *E. coli* BL21 DE3. Bacterial cells were grown to log phase at 37°C in Terrific Broth (1.2% Polypeptone peptone, 2.4% Yeast Extract, 0.6% LB Broth, 0.4% glycerol, 17 mM $KH_2PO_4$, 72 mM $K_2HPO_4$), cooled on ice, and the

expression of transgene were induced with 0.1 mM IPTG for 18–20 h at 18°C. For NMR spectroscopy, $^{15}N$-labeled or $^{13}C/^{15}N$-labeled INCENP fragments were expressed in M9 minimal medium containing $^{15}N$-ammonium chloride with or without $^{13}C$-glucose.

For GST fusion fragments in Fig. S2 D and Fig. S3 A (except for INCENP 121–178), bacterial cells were harvested by centrifugation at 6,500 rpm for 10 min, and cell pellets were resuspended in Buffer A (20 mM Tris at pH 8.0, 500 mM NaCl, 10 mM imidazole), supplemented with a cocktail of protease inhibitors (Complete EDTA-free; Roche). Cells were then ruptured by sonication on ice and the insoluble fraction was removed twice by centrifugation at 4,500 rpm for 5–10 min at 4°C. Supernatants were incubated with Ni-NTA Agarose HP (Wako) in a rotating tube for 1.5 h at 4°C. After washing with Buffer A, the tagged protein was eluted with Buffer A containing 500 mM imidazole. GST-tag was removed by adding (GST-tagged) PreScission protease and then dialyzed twice against Buffer B (50 mM Tris at pH 8.0, 300 mM NaCl, 0.5 mM EDTA, 1 mM DTT) for 2 and 12 h at 4°C. PreScission protease and GST-tag were removed by Glutathione Sepharose 4B (GE Healthcare) column and the flow-through fraction was concentrated with Amicon Ultra (Millipore), and further purified by size exclusion chromatography (HiLoad 16/600 Superdex 200 prep grade) with Buffer C (20 mM NaPO₄ at pH 7.0, 150 mM NaCl, 1 mM DTT). The eluted fractions were collected and frozen at –80°C until further usage. Custom-made peptide of hINCENP (159–178) was purchased from BEX Company Limited (Fig. 2 D).

For GST fusion of Tyr-added fragments (Fig. S2 D [121–178 fragment]; Fig. S3 C and Fig. S5 A), bacterial pellets were resuspended by Buffer D (20 mM Tris at pH 8.0, 500 mM NaCl, 1 mM DTT) and supplemented with a cocktail of protease inhibitors (Complete EDTA-free). Cells were ruptured by sonication on ice and supplemented with Polyethylenimine P-70 Solution (final conc. 0.1%) (Wako), followed by vortexing and standing on ice for 10 min. The insoluble fraction was removed twice by centrifugation, and the soluble fraction was bound to Glutathione Sepharose 4B using Econo-Column (2.5 × 20 cm; Bio-Rad). Being washed with Buffer D, the tagged protein was eluted with Buffer E (150 mM Tris at pH 9.2, 500 mM NaCl, 1 mM DTT, 50 mM reduced glutathione). Cleavage of the GST-tag by PreScission was proceeded during dialysis with Buffer B for 12 h at 4°C.

For INCENP 121–178 fragment (Fig. S2 D), the purified protein was dialyzed with Buffer B (-EDTA, +10 mM imidazole). The sample was loaded onto the Glutathione Sepharose 4B to eliminate surplus GST-tagged proteins and further purified by Ni-NTA HP column chromatography (GE Healthcare). The binding fraction was eluted, collected, concentrated, and further purified by size exclusion chromatography with Buffer C.

For other proteins, GST-tag cleaved samples were dialyzed twice with low salt Buffer F (20 mM NaPO₄ at pH 7.0, 75 mM NaCl, 1 mM DTT) for 2 h at 4°C, loaded onto the Glutathione Sepharose 4B to eliminate surplus GST-tagged proteins, and further purified by HiTrap Q HP column chromatography (GE Healthcare). The flow-through fraction was collected,

concentrated, and further purified by size exclusion chromatography with Buffer C. GST fusion Tyr-added hINCENP fragment (160–210) and GST fusion Tyr-added hSgo1 fragment (405–494) used in Fig. 6 A were purified by size exclusion chromatography with Buffer C directly after the first dialysis. The resulting elution fractions were concentrated and stocked at −80°C. Although the INCENP 160–192 fragment was revealed to be a smeared band in SDS-PAGE and CBB staining (Fig. S3 C), it consisted of a single molecular weight of INCENP fragment (160-192), confirmed by mass spectrometry MALDI-TOF Autoflex (Bruker Daltonis). To purify GST-tagged (non-cleaved version of) INCENP fragments (Fig. 2 B), size exclusion chromatography with Buffer C was used.

For recombinant HP1α proteins (Fig. S2 C and Fig. S3 B), a crude fraction of 6His-MBP-tagged full-length hHP1α WT was prepared following the protocol for Tyr-fusion INCENP fragments, except for the use of Amylose Resin (New England Biolab) and 10 mM maltose for elution in Buffer D-base. After the removal of 6His-MBP-tag by 6His-tagged TEV protease during dialysis with Buffer B, followed by Buffer D exchange twice, the sample was loaded onto the Amylose Resin column to remove the tag. The flow-through fraction was then cleared of 6His-TEV protease and surplus 6His-MBP-tag by Ni-NTA Agarose HP (Wako), supplemented with 20 mM imidazole to prevent nonspecific binding of HP1α to the beads. The flow-through fraction was dialyzed twice with Buffer F for 2 and 12 h at 4°C, and was further purified by HiTrap Q HP column chromatography with a linear gradient of 75–1,000 mM NaCl. To obtain HP1α CSD used in NMR spectroscopy, a crude fraction of 6His-tagged hHP1α CSD was prepared using Ni-NTA Agarose HP (Wako) and 500 mM imidazole in Buffer A for elution. TEV-mediated removal of 6His-tag during dialysis with Buffer B was followed by dialysis with Buffer A, and 6His-TEV protease was removed by Ni-NTA Agarose HP. The elution fractions were concentrated and further purified by size exclusion chromatography with Buffer C.

## Pull-down assay

10 μM of full-length hHP1α WT was mixed with 5 μM of a series of GST-tagged hINCENP -6His fragments in a total of 500 μl of binding buffer (20 mM NaPO₄ at pH 7.0, 150 mM NaCl, 1 mM DTT) and incubated for 30 min at 4°C. The mixture solution was then incubated with 100 μl slurry of Glutathione Sepharose 4B resin and then incubated for another 30 min at 4°C. The beads were washed three times with the binding buffer, and the resulting precipitates were analyzed by CBB staining.

## Isothermal titration calorimetry (ITC)

Protein solution (10 μM, hINCENP peptide or recombinant fragments) was loaded into a 1.4 ml-cell of VP-ITC isothermal titration calorimeter (Microcal) (Fig. 2, C and D; and Fig. 3, D and E). The solution was titrated against 100 μM ligand solution (hHP1α full-length or CSD in a dimer) via a 250 μl titration syringe iteratively with 25 steps of 10 μl each. Experiments were carried out at 20°C. ITC analysis in Fig. 6 A and Fig. S5 A was measured using the iTC200 isothermal titration calorimeter (Malvern). Protein solution (hHP1α full-length was prepared at

the concentration of 60 μM, as a dimer, for hINCENP titration and 80 μM for hSgo1 titration) was loaded into a 200-μl cell of iTC200. The solution was titrated against 600 μM ligand (hINCENP) or 800 μM ligand (hSgo1) via a 40-μl titration syringe iteratively with 19 steps of 2 μl each. Experiments were carried out at 25°C. The ligand solution was prepared in the same buffer as the protein (Fig. 2 C: 10 mM Tris at pH 8.0, 150 mM NaCl, 1 mM DTT; Fig. 2 D, Fig. 3, D and E, Fig. 6 A, and Fig. S5 A: 20 mM NaPO₄ at pH 7.0, 150 mM NaCl, 1 mM DTT). The heat of dilution generated by the ligand was subtracted, and the binding isotherms were fitted to a one-site binding model by using Origin 7 software (Microcal).

## Nuclear magnetic resonance (NMR) spectroscopy

The protein concentrations were 0.1–1 mM in 20 mM NaPO₄ at pH 7.0, 150 mM NaCl, 1 mM DTT, and 5% D₂O. NMR experiments were performed on AVANCE 600-MHz and AVANCE III HD 950-MHz spectrometers with a triple-resonance TCI cryogenic probe (Bruker Bio Spin) at 298 K.

For backbone assignments, the spectra of HNCO, HN(CA)CO, HNCA, HN(CO)CA, HNCACB, and HN(CO)CACB were measured for sequential assignments of the backbone $^{1}$H, $^{13}$C, and $^{15}$N chemical shifts of HP1α_CSD and INCENP fragments (160-210 or 160-192). NMR data were processed by NMRPipe (Delaglio et al., 1995), and signal assignments were performed with Magro (Kobayashi et al., 2007). NMR data were analyzed by NMRViewJ (One Moon Scientific), and PINT. The chemical shift differences Δδ were calculated by the equation $\Delta\delta = (\Delta\delta_H) + (\Delta\delta_N/5)^2$, where $\Delta\delta_H$ and $\Delta\delta_N$ are chemical shift differences of the amide proton and nitrogen atoms, respectively.

In NMR titration experiments, isotopically labeled INCENP proteins ($^{13}$C/$^{15}$N labeled or $^{15}$N labeled) were titrated by gradual addition of unlabeled HP1α to the NMR tube; the concentrated HP1α solution was added step by step to the INCENP solution until the free INCENP signals disappear. The INCENP solution was added to the isotopically labeled CSD dimer ($^{13}$C/$^{15}$N labeled or $^{15}$N labeled) solution until the free CSD signals disappeared and the complex solution was then concentrated for NMR measurements using Amicon Ultra (Millipore). The $^{1}$H–$^{15}$N HSQC spectrum of the INCENP-HP1α_CSD mixture was acquired at 298 K.

The intermolecular NOEs between INCENP (160-192) and HP1α_CSD dimer were obtained from 3D $^{13}$C/$^{15}$N-filtered $^{15}$N -edited NOESY (120 ms mixing time) and 3D $^{15}$N -edited NOESY (120 ms mixing time) spectra (Breeze, 2000; Schleucher et al., 1994).

## Deposited data of NMR spectroscopy

We deposited the above NMR data at the Biological Magnetic Resonance Data Bank entry. The accession numbers are #52192 and #52194.

## Docking model

Docking calculations between HP1α_CSD (111-179) and INCENP (160-192) were carried out using the HADDOCK 2.2 web server (de Vries et al., 2010; Honorato et al., 2021). In these docking calculations, the initial structure of HP1α_CSD (111-179) was

obtained using MODELLER (Martí-Renom et al., 2000) based on the X-ray crystal structure (PDB code: 3I3C), which consists of 111–173 amino acids with missing residues of 132–134. The initial structure of INCENP (160-192) was derived using the CNS 1.2 (Brünger et al., 1998), incorporating 38 dihedral angles (phi and psi angles for amino acids 165–171, 173, 174, and 177–186) from TALOS analysis (Shen et al., 2009) and six hydrogen bonds between residues 181/177, 182/178, 183/179, 184/180, 185/181, and 186/182 from the main chain chemical shift of the complex, which showed a β-strand (164-171) and an α-helix (177-186). Ambiguous restraints for the docking calculations were classified into two zones based on the complex structure of HP1β_CSD and Sgo1 (PDB code: 3Q6S), in which a β-sheet was formed between HP1β_CSD with anti-parallel (168–173) and parallel (168–170) and Sgo1 (448–455) corresponding to HP1α_CSD with anti-parallel (172–177) and parallel (172–174) and INCENP (164–171). The first zone, where a β-sheet is anticipated to form between HP1α_CSD with anti-parallel (172–177) and parallel (172–174) and INCENP (164–171), had active residues for HP1α identified from significant chemical shift perturbation 0.5 ppm as Ala129, Asp131, Glu169, Arg171, Leu172, His175, Ala176, Tyr177, and for INCENP as Pro167, Val168, Val169, Glu170, Ile171. In the second zone outside the β-sheet formed between HP1α_CSD with anti-parallel (172-177) and parallel (172–174) and INCENP (164–171), active residues showing significant chemical shift perturbations over 0.3 ppm were identified as Ile113, Arg115, Phe117, Trp142 for HP1α and Glu180, Gln181, His182, Val183 for INCENP.

In the docking calculations, initially, 1,000 structures were generated using rigid molecular docking. The 200 structures with the lowest energy were first refined by simulated annealing in vacuum, followed by further simulated annealing in a water molecule shell. This refinement of the 200 structures incorporated dihedral angles and hydrogen bond restraints from TALOS analysis. Additionally, distance restraints within 5 Å were introduced for N, C′, O, CA, and CB of HP1α_CSD -INCENP corresponding to residues forming β-sheets in the HP1β_CSD-Sgo1 complex. When clustering was performed on the 200 obtained structures based on the position of the helix (177–186) of INCENP in the complex with an RMSD threshold of 2.5 Å, 72 out of the 200 structures were classified into eight clusters. Among these, the cluster with the most structures and the highest HADDOCK score, consisting of 22 structures, was selected as the final model.

### Quantifications of fluorescence intensities

The quantitation of immunofluorescence intensities was performed using Fiji software (National Institutes of Health) (Fig. 5, C and E; and Fig. 6 H). To quantify the HP1α foci, a circle with a diameter of six pixels was centered for each focus, and pixel intensity within the circle was measured in unprocessed images. The average intensity of randomly selected 10 points on the chromosomal area was used as a background intensity. In Fig. 5 C, Myc foci were used to define ROI, where Myc and HP1α intensities were measured simultaneously (i.e., the two signal intensities were acquired from the same region). After background subtraction from each signal intensity, ratios of HP1α to

Myc intensities were calculated. In Fig. 5 E and Fig. 6 H, a similar procedure was applied in normalizing Hec1-pS44 signal to Hec1.

In Fig. 6, C and E, the foci of HP1α were semiautomatically detected using CellPathfinder software (Yokogawa). Within the area of manually defined DAPI-positive nuclei, Myc foci above the threshold (Detect Factor: 0.9, Granule Diameter: 3 μm) were detected, and fluorescence intensities of Myc and HP1α at each focus were simultaneously measured. The background region was automatically decided from the nuclear area excluding the Myc-foci. After background subtraction from each signal intensity, signals of HP1α were normalized to Myc. The resulting quantifications were presented in beeswarm plots depicted in Prism (Graphpad). Statistical analyses were conducted as described in the figure legends.

### Online supplemental material

Fig. S1 shows centromeric localization of HP1α, INCENP, and Sgo1. Fig. S2 shows INCENP–HP1 interaction metrics based on ITC measurements. Fig. S3 shows extended analyses of the INCENP and HP1 interaction. Fig. S4 shows the importance of the SSH domain in the mitotic function of the CPC. Fig. S5 shows α-helical segment of the SSH domain supports stable interaction with HP1.

### Data availability

All unique reagents and materials reported in this study will be made available upon reasonable request without restrictions. All data reported in this paper will be shared by the corresponding author upon request. Any additional information required to reanalyze the data reported in this paper is available from the corresponding author upon request.

## Acknowledgments

We are grateful to Ana Losada for Sgo1 and HP1α antibodies, Jennifer G. DeLuca for phosphor-Hec1 antibodies, Tatsuya Nishino and Hideaki Ohtomo for technical help with recombinant protein purification and ITC experiments, Utako Kato for handling cell-culture experiments, Kazuhiko Tomita for helping quantification analysis of CQ1, and Minji Jo, Norihisa Shindo, and Yusuke Abe for intellectual suggestions.

Research in the T. Hirota lab is supported by the Japan Society for the Promotion of Science (JSPS) Grant-in-Aid for Scientific Research (22H04996, 22H00458, 18H04034 [to T. Hirota], 22K15043, 19K16047 [to K. Sako] and 22H05608, 22K19465, 20H03190 [to R-S. Nozawa]). Research in the Y. Nishimura lab is supported, in part, by NMR Platform (grant no. JPMXS0450100021 to Y. Nishimura) from the Ministry of Education, Culture, Sports, Science and Technology (MEXT), Japan; by a Platform Project for Supporting Drug Discovery and Life Science Research (Basis for Supporting Innovative Drug Discovery and Life Science Research; BINDS) from the Japan Agency for Medical Research and Development (AMED; grant nos. JP21am0101073 and JP22ama121001 to Y. Nishimura); and by JSPS Grants-in-Aid for Scientific Research (JP23H02426 to Y. Nishimura). K. Sako acknowledges the support from the JSPS Fellowship (PD).

Author contributions: K. Sako: Conceptualization, Data curation, Formal analysis, Funding acquisition, Investigation, Methodology, Project administration, Resources, Validation, Visualization, Writing—original draft, Writing—review & editing, A. Furukawa: Data curation, Investigation, Visualization, Writing—review & editing, R-S. Nozawa: Investigation, Methodology, Resources, Writing—review & editing, JI. Kurita: Formal analysis, Software, Validation, Writing—review & editing, Y. Nishimura: Conceptualization, Funding acquisition, Project administration, Supervision, Validation, Writing—review & editing, T. Hirota: Conceptualization, Funding acquisition, Project administration, Supervision, Writing—review & editing.

Disclosures: The authors declare no competing interests exist.

Submitted: 5 December 2023

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

# Supplemental material

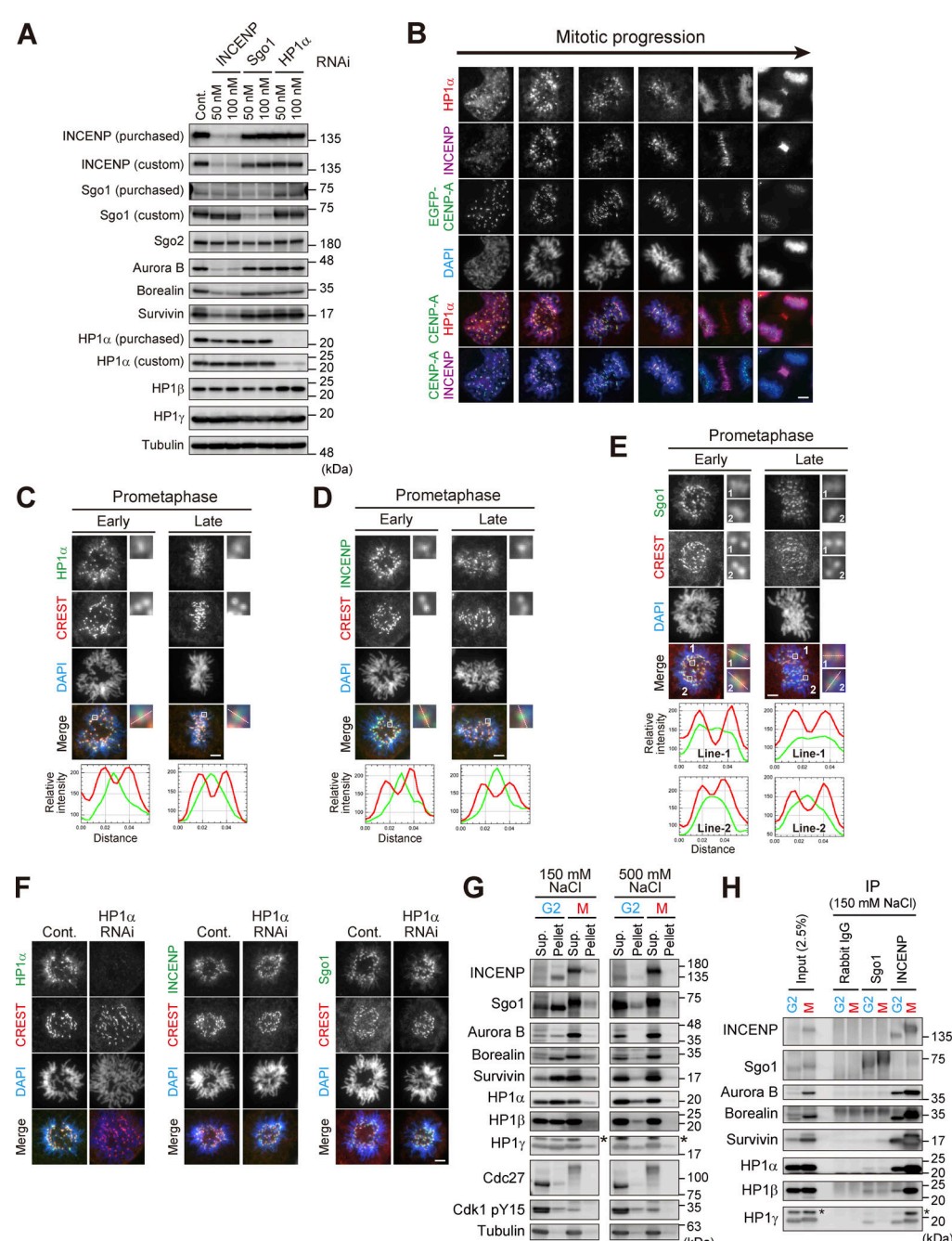

Figure S1.    **Centromeric localization of HP1α, INCENP, and Sgo1. (A)** Verification of the effect of siRNA and specificity of antibodies. HeLa cells were transfected with a mock (Cont.) or siRNA at indicated concentrations, during a 24 h of thymidine synchronization (for INCENP and Sgo1) or even 24 h before the synchronization (for HP1α), and mitotic cells were enriched by treating cells with 7.5 μM STLC for 15 h. Samples were immunoblotted with the indicated antibodies. Note that the final concentration of 50 nM of siRNA to HeLa cells can sufficiently knock down the targeted proteins and therefore used throughout this study. **(B)** Immunofluorescence microscopy of chromosome spreads prepared from unperturbed mitotic HeLa cells expressing EGFP-CENP-A (see Materials and methods for details), which allowed for unambiguous detection of centromere proteins under the condition in which spatial arrangements and mitotic chromosome morphologies were overall preserved. These samples were stained with antibodies to HP1α and INCENP. Scale bar, 5 μm. **(C–E)** Immunofluorescence microscopy of chromosome spreads prepared from unperturbed mitotic HeLa cells as in B. These cells were first stained with CREST serum and next stained with antibodies to HP1α (C), INCENP (D), and Sgo1 (E), respectively. Note that during prometaphase progression, the line scan profile of HP1α was consistently similar to that of INCENP, i.e., single focused peak signal at inner centromeres; whereas that of Sgo1 was rather broad and often ranged throughout the centromere. Scale bar, 5 μm. **(F)** Immunofluorescence microscopy of chromosome spreads prepared from unperturbed mitotic HeLa cells as in B. HeLa cells treated with or without siRNA to HP1α were first stained with CREST serum and then with antibodies to HP1α (left panels), INCENP (middle panels), and Sgo1 (right panels), respectively. Scale bar, 5 μm. **(G)** Immunoprecipitation with different salt concentrations. Cell extraction with a buffer containing 500 mM NaCl solubilized chromatin-related proteins more efficiently than 150 mM NaCl in both G2 and M phases. Cdk1-pY15 is used as a marker of G2 phase. Asterisks: non-specific crossed species. **(H)** Immunoprecipitation with indicated antibodies from cell lysate prepared with 150 mM NaCl was immunoblotted with the indicated antibodies. In Sgo1-IP, the HP1 signals were barely detectable even in this low stringency condition. Asterisks: non-specific crossed species. Source data are available for this figure: SourceData FS1.

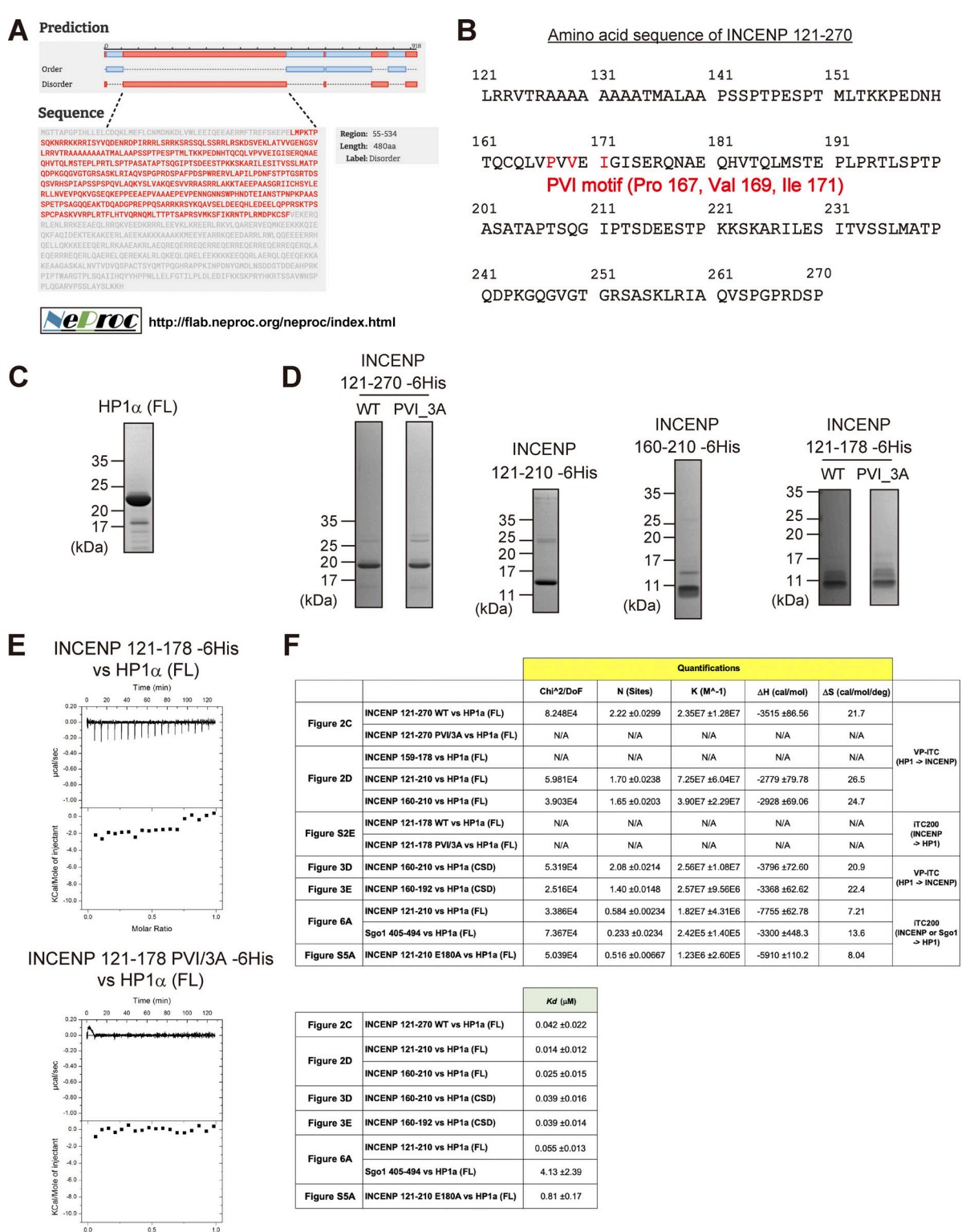

Figure S2. **INCENP-HP1 interaction metrics based on ITC measurements. (A)** Large part of INCENP consists of intrinsically disordered regions. The predictions were executed by a website of the NeProc (Next ProS Classifier). **(B)** The amino acid sequence of INCENP 121–270 containing the PVI motif. **(C and D)** Purified recombinant proteins used in the binding assay (HP1α, Fig. 2 B), ITC assay (Fig. 2, C and D, Fig. 3 D, Fig. 6 A, and Fig. S5 A), and NMR spectroscopy (Fig. 3, B and C). Coomassie brilliant blue staining. **(E)** The ITC measurements of the binding between INCENP fragments (121–178, wild type or PVI_3A mutant) and HP1α dimer (full-length). Note that both fragments fail to specifically interact with HP1. **(F)** Quantifications in the ITC analyses. Note that all the INCENP fragments showed similar degree of dissociation constant Kd to HP1α (0.01–0.04 μM) when they contained the PVI motif and the C-terminally juxtaposed domain (Related to Fig. 2, C and D, Fig. 3, D and E; and Fig. S2 E). The Sgo1 fragment with the canonical PVI motif showed much lower Kd (~4 μM) compared with the INCENP fragment (Related to Fig. 6 A). The INCENP 121–210 E180A fragment, having the intact PVI motif and the mutated α-helical region of the SSH domain, showed much larger Kd (~0.8 μM) compared with the WT (related to Fig. S5 A). Source data are available for this figure: SourceData FS2.

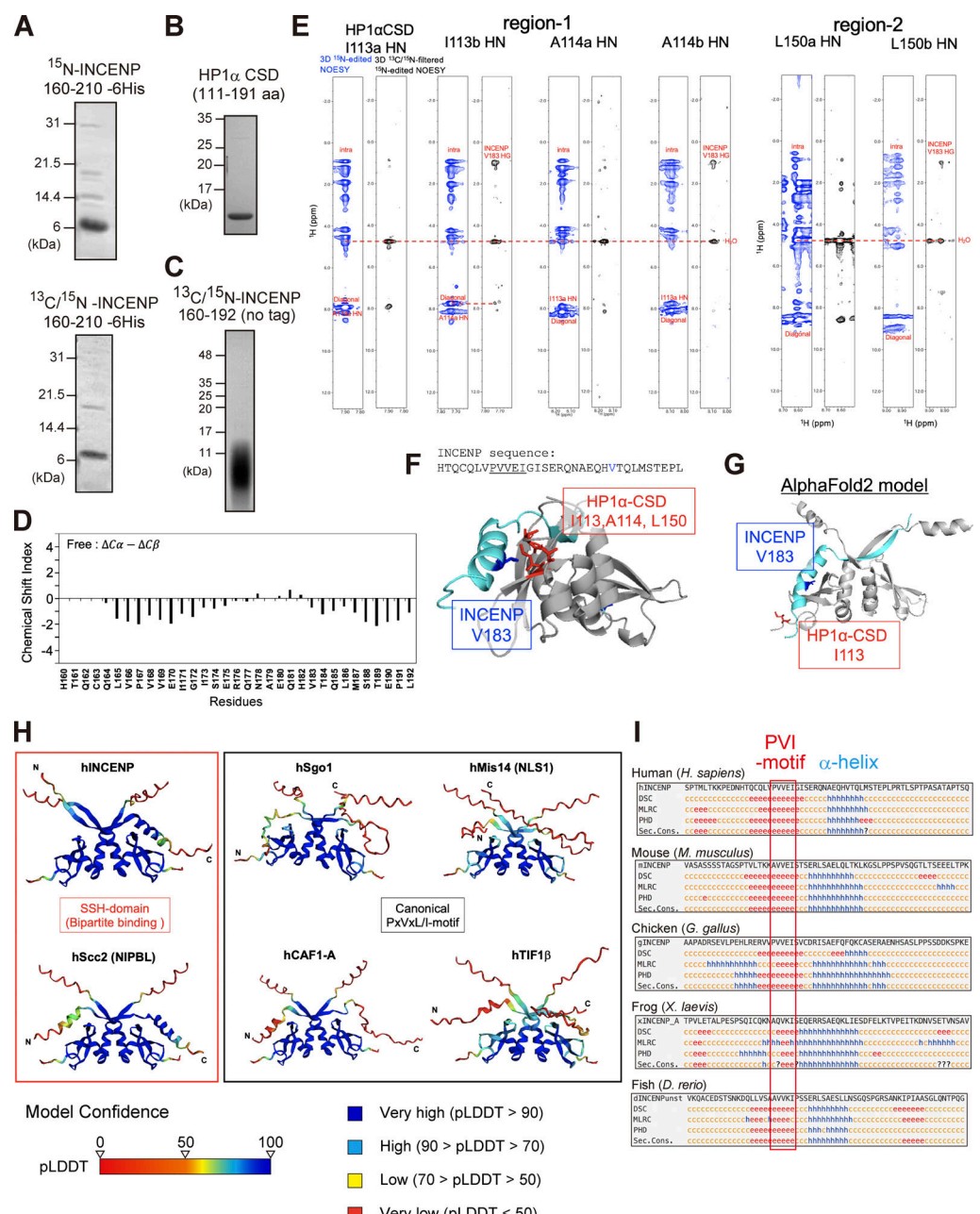

Figure S3. **Extended analyses of the INCENP and HP1 interaction. (A and B)** Purified recombinant proteins used in the $^1$H-$^{15}$N heteronuclear single quantum coherence (HSQC) NMR analysis. $^{15}$N-INCENP (160–210) -6His used in Fig. 3, A and B, $^{13}$C/$^{15}$N-INCENP (160–210) -6His used in Fig. 3 A, HP1α chromoshadow domain (CSD) used in Fig. 3 are shown. Coomassie brilliant blue staining. **(C)** Purified recombinant protein of $^{13}$C/$^{15}$N-INCENP (160–192) (untagged version) used in the HSQC NMR analysis shown in Fig. 3, F–H; and Fig. S3 D. Coomassie brilliant blue staining. **(D)** Chemical shift indices (CSI) of INCENP (160–192) alone were calculated from $^{13}$C$_α$ and $^{13}$C$_β$ (black). **(E)** Alternating strips of the 3D $^{15}$N-edited NOESY and 3D $^{13}$C/$^{15}$N-filtered $^{15}$N-edited NOESY for residues Ile113 and Ala114 within HP1α CSD dimer. The intermolecular NOEs between INCENP (160–192) and HP1α CSD dimer were obtained from 3D $^{13}$C/$^{15}$N-filtered $^{15}$N-edited NOESY (120 ms mixing time) and 3D $^{15}$N-edited NOESY (120 ms mixing time) spectra. **(F)** A docking model between HP1α_CSD (111–179 aa) dimer and INCENP (160–192 aa) using the HADDOCK depicted in Fig. 5 A. Although we have not used the NOE results to obtain this docking model (Fig. S3 E), it could nevertheless explain that the strong and weak intermolecular NOE signals from amide protons of Ile113, Ala114, and Leu150 of one subunit of the CSD dimer due to two methyl groups of Val183 of INCENP, exemplifying the reliability of the model. **(G)** The prediction model of AlphaFold2 between INCENP (160–192 aa) and HP1α CSD (111–191 aa) dimer. Note that the amide protons of Ile113 of both subunits in the CSD dimer are far from the methyl groups of Val183 of INCENP, which exemplifies the difference between the observation and the prediction. **(H)** The complex prediction model of AlphaFold2 between each HP1 interactor's fragment and HP1α CSD (110–191 aa) dimer. The fragments used were hINCENP 148–209 aa, hScc2 982–1,043 aa, hSgo1 432–493 aa, hMis14 (NLS1) 190–251 aa, hCAF1-A 201–262 aa, and hTIF1β 467–528 aa. The model confidence score (pLDDT; predicted Local Distance Difference Test), a per-residue model produced by AlphaFold2, is indicated in different colors between 0 and 100. Region with a pLDDT below 50 may be unstructured in isolation. **(I)** Secondary structure predictions (Sec. Cons.) were based on several algorithms including DSC, MLRC, PHD (PRABI-GERLAND, NPS@ Secondary Consensus Structure Prediction). INCENP amino acid sequences of the mouse (136–197), chicken (136–197), frog (135–196), and fish (165–226) including possible PVI motifs were aligned for comparison with the human (148–209). Source data are available for this figure: SourceData FS3.

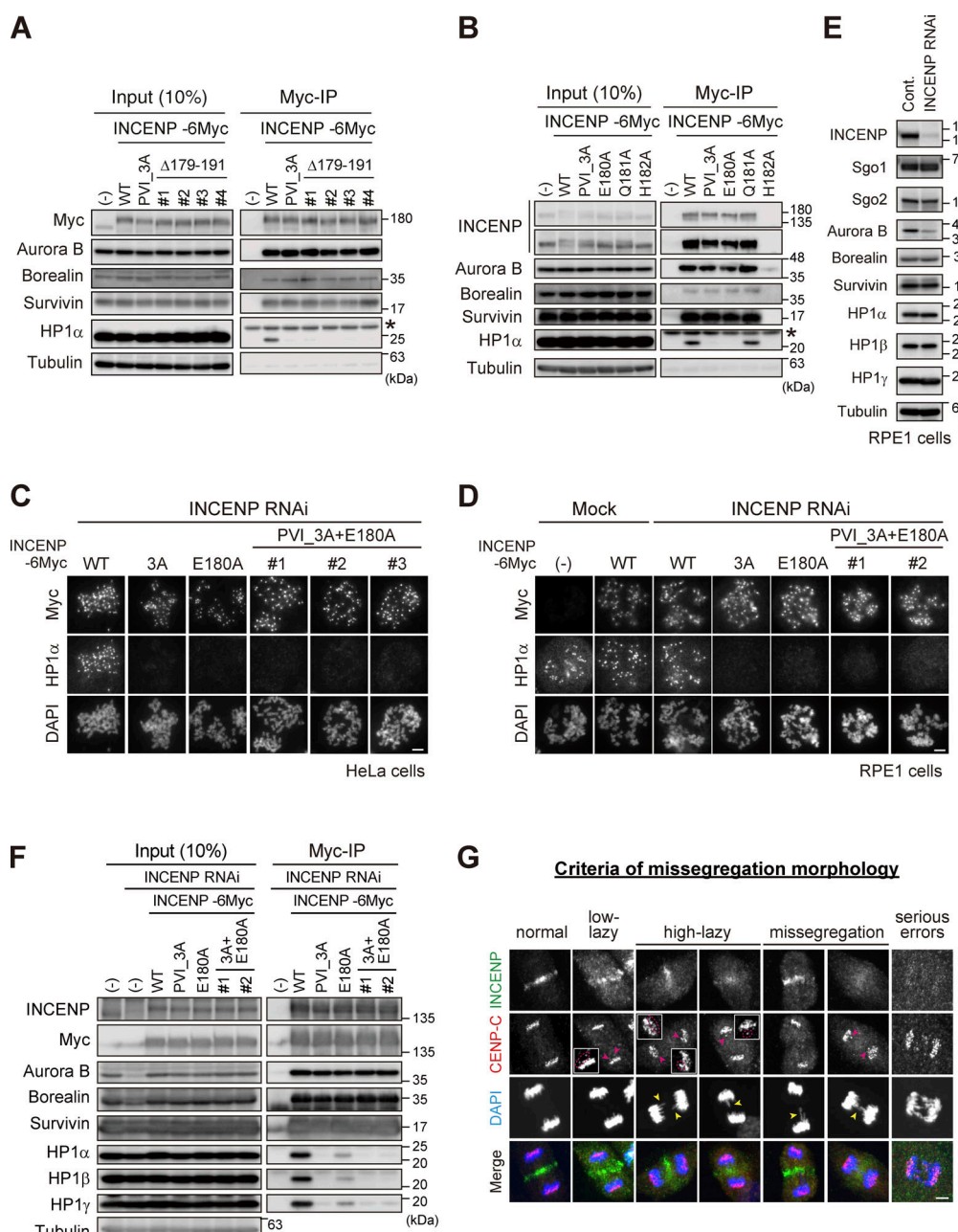

Figure S4. **Importance of the SSH domain in the mitotic function of the CPC.** **(A)** Immunoprecipitation with myc antibodies from mitotic extract prepared from parental, WT, PVI_3A, Δ179–191 INCENP-expressing HeLa cells, and immunoblotted with the indicated antibodies. (−) indicates the parental cells. Note that all tested clones of INCENP Δ179-191 mutant, which involves amino acids folding the α-helix, showed a similar effect to the PVI_3A mutant (clone #1 is used in Fig. 5 B). An asterisk indicates non-specific proteins (light chain of IgG). **(B)** Immunoprecipitation with myc antibodies from mitotic extract prepared from parental, WT, PVI_3A, E180A, Q181A, H182A INCENP-transient expressing HeLa cells, and immunoblotted with the indicated antibodies. INCENP H182A mutant did not express in cells for unknown reasons. An asterisk indicates non-specific proteins (light chain of IgG). **(C and D)** Immunofluorescence microscopy of chromosome spreads from indicating cell lines of HeLa (C) or RPE1 (D). Cells were transfected with a mock or siRNA to INCENP for 48 h, followed by 100 ng/ml nocodazole for 2 h (HeLa) or 150 ng/ml nocodazole for 12 h (RPE1), and were fixed and stained with antibodies to myc and HP1α. Note that all INCENP PVI_3A+E180A clones revealed a similar effect on HP1α to that of PVI_3A and E180A mutants. Scale bar, 5 μm. **(E)** Verification of siRNA to INCENP in RPE1 cells. Cells were transfected with a mock (Cont.) or 20 nM siRNA during synchronization with palbociclib treatment for 24 h, and mitotic cells were enriched by treating cells with 7.5 μM STLC for 15 h. Samples were immunoblotted with the indicated antibodies. **(F)** Immunoprecipitation with myc antibodies from mitotic extract prepared from the parental and indicated version of INCENP-expressing RPE1 cells and immunoblotted with the indicated antibodies. Cells were treated with siRNA to INCENP following the protocol described in E. **(G)** Anaphase cells stained with antibodies to INCENP and CENP-C were classified into five groups based on the degree of lagging chromosomes as described (Sen et al., 2021): normal, when cells have no detectable delay in separating chromosomes and centromeres; low-lazy, when cells are positive for lagging centromeres, without detectable lagging of chromosomes; high-lazy, when cells are positive for lagging centromeres associated with lagging chromosomes whose arms are protruded from the separating chromosomes mass; missegregation, when cells are positive for typical lagging and bridge chromosomes; serious errors, when cells are positive for massive chromosome missegregation. Scale bar, 5 μm. Source data are available for this figure: SourceData FS4.

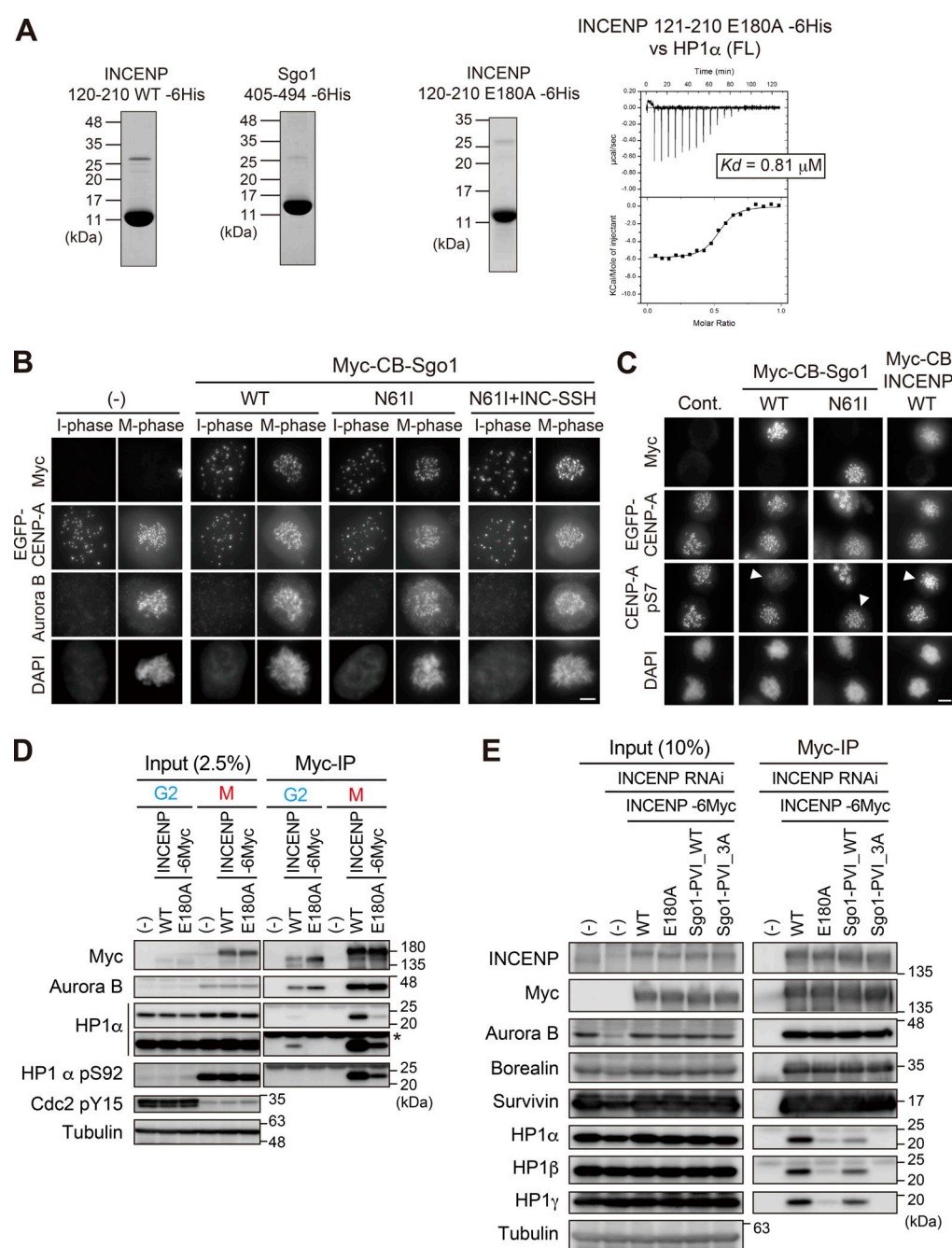

Figure S5.  **α-Helical segment of SSH domain supports stable interaction with HP1. (A)** Left panels: Purified recombinant proteins used in ITC assay in Fig. 6 A (INCENP 121–210 WT and Sgo1 405–494) and Fig. S5 A (INCENP 121–210 E180A). Coomassie brilliant blue staining. Right panel: The ITC measurements of the binding between INCENP fragment (121–210 E180A mutant) and HP1α dimer (full-length). **(B)** Immunofluorescence microscopy of HeLa cells transiently expressing the Myc-CB fused Sgo1 mutants. Fixed cells were stained with antibodies to Aurora B and myc. The fluorescence of EGFP-CENP-A is used as a reference which is constitutively expressed in these cells. Note that over-expression of Sgo1 constructs at centromeres had little effect on the localization of Aurora B, in agreement with the previous study (Meppelink et al., 2015). Scale bar, 5 µm. **(C)** Immunofluorescence microscopy of HeLa cells transiently expressing the Myc-CB fused Sgo1 or INCENP mutants. Fixed cells were stained with antibodies to myc-tag and CENP-A-pS7, and constitutively expressed EGFP-CENP-A is used as a reference. Arrowheads indicate cells expressing CB-fusion protein (myc positive cells). Note that over expression of Myc-CB fused Sgo1 wild-type (WT) in centromeres reduced the level of Aurora B-mediated Ser7 phosphorylation of CENP-A, a readout for Aurora B activity. Whereas over expression of Myc-CB fused Sgo1 N61I (PP2A binding deficient mutant) and INCENP doubleΔ mutant had little effect on CENP-A-pS7. These data suggest that the ectopic recruitment of PP2A via Sgo1 WT antagonize the Aurora B activity, in agreement with the previous study (Meppelink et al., 2015). Scale bar, 5 µm. **(D)** Binding of HP1α to INCENP in G2 phase depends also on the α-helix of the SSH domain. Immunoprecipitation with myc antibodies from indicated HeLa cell extracts in the G2 phase (10 µM RO3306 treatment for 9 h) or M phase (7.5 µM STLC treatment for 16 h), followed by immunoblotting with the indicated antibodies. Cdk1 pY15 and HP1α pS92 are markers of the G2 phase and M phase, respectively. (–) indicates parental cells. An asterisk indicates non-specific IgG light chain. **(E)** Immunoprecipitation with myc antibodies from mitotic extract prepared from RPE1 cells that express the indicated version of INCENP in place of endogenous INCENP, and immunoblotted with the indicated antibodies. Mitotic cells were enriched as described in Fig. S4 E. Source data are available for this figure: SourceData FS5.

