## [Peer Review File · The Journal of Cell Biology]

Bipartite binding interface recruiting HP1 to chromosomal passenger complex at inner centromeres

Kosuke Sako, Ayako Furukawa, Ryu-Suke Nozawa, Jun-ichi Kurita, Yoshifumi Nishimura, and Toru Hirota

Corresponding Author(s): Toru Hirota, Japanese Foundation For Cancer Research and Yoshifumi Nishimura, Yokohama City University

Review Timeline:

Submission Date:	2023-12-05
Editorial Decision:	2024-01-26
Revision Received:	2024-04-05
Editorial Decision:	2024-04-24
Revision Received:	2024-05-02

Monitoring Editor: Karen Oegema

Scientific Editor: Andrea Marat

Transaction Report:

DOI: <https://doi.org/10.1083/jcb.202312021>

January 26, 2024

Re: JCB manuscript #202312021

Dr. Toru Hirota
Japanese Foundation For Cancer Research
Ariake 3-8-31
Tokyo, Tokyo 135-8550
Japan

Dear Dr. Hirota,

Thank you for submitting your manuscript entitled "Bipartite binding interface recruiting HP1 to chromosomal passenger complex at inner centromeres". The manuscript was assessed by expert reviewers, whose comments are appended to this letter. We invite you to submit a revision if you can address the reviewers' key concerns, as outlined here.

You will see that the reviewers find that your study provides interesting insight into our understanding of CPC regulation, and as such we agree a suitably revised study is of interest to the readership of JCB. The reviewers have provided constructive feedback, and we agree it will be important to address all of the main points raised by Reviewers 1 and 2.

GENERAL GUIDELINES:

Text limits: Character count for an Article is < 40,000, not including spaces. Count includes title page, abstract, introduction, results, discussion, and acknowledgments. Count does not include materials and methods, figure legends, references, tables, or supplemental legends.

Figures: Articles may have up to 10 main text figures. Figures must be prepared according to the policies outlined in our Instructions to Authors, under Data Presentation, <https://jcb.rupress.org/site/misc/ifora.xhtml>. All figures in accepted manuscripts will be screened prior to publication.

Supplemental information: There are strict limits on the allowable amount of supplemental data. Articles may have up to 5 supplemental figures. Up to 10 supplemental videos or flash animations are allowed. A summary of all supplemental material should appear at the end of the Materials and methods section.

Please note that JCB now requires authors to submit Source Data used to generate figures containing gels and Western blots with all revised manuscripts. This Source Data consists of fully uncropped and unprocessed images for each gel/blot displayed in the main and supplemental figures. Since your paper includes cropped gel and/or blot images, please be sure to provide one Source Data file for each figure that contains gels and/or blots along with your revised manuscript files. File names for Source Data figures should be alphanumeric without any spaces or special characters (i.e., SourceDataF#, where F# refers to the associated main figure number or SourceDataFS# for those associated with Supplementary figures). The lanes of the gels/blots should be labeled as they are in the associated figure, the place where cropping was applied should be marked (with a box), and molecular weight/size standards should be labeled wherever possible.

The typical timeframe for revisions is three to four months. While most universities and institutes have reopened labs and allowed researchers to begin working at nearly pre-pandemic levels, we at JCB realize that the lingering effects of the COVID-19 pandemic may still be impacting some aspects of your work, including the acquisition of equipment and reagents. Therefore, if you anticipate any difficulties in meeting this aforementioned revision time limit, please contact us and we can work with you to find an appropriate time frame for resubmission. Please note that papers are generally considered through only one revision

cycle, so any revised manuscript will likely be either accepted or rejected.

Thank you for this interesting contribution to Journal of Cell Biology. You can contact us at the journal office with any questions at cellbio@rockefeller.edu.

Sincerely,

Karen Oegema, PhD
Monitoring Editor

Andrea L. Marat, PhD
Senior Scientific Editor

Journal of Cell Biology

Reviewer #1 (Comments to the Authors (Required)):

Previous studies from the Hongtao Yu (Kang et al., 2011) and Toru Hirota (Abe et al., 2016) labs showed that INCENP possesses an HP1 binding PxVxl motif and INCENP-HP1 interaction mediated by this motif is critical to achieve the full activity of the Aurora B kinase and its effective error-correction role. This manuscript by Sako et al., through detailed structural (NMR and computational modelling) and biochemical characterisation, demonstrates that INCENP-HP1 binding involves not only the PxVxl motif but also a downstream alpha-helical segment. Disrupting the alpha-helix interaction with HP1 (using INCENP E180A mutation) reduced HP1 inner centromere localisation to the same extent as the PxVxl mutant. Consistent with this, the Aurora B activity, as assessed by the Hec1 phosphorylation, was also perturbed by both PxVxl and E180A mutants. Based on these observations, the authors claim that the bipartite mode of INCENP binding to HP1 is crucial for the inner centromere localization of HP1 and the full activity of the CPC. By ectopically tethering different INCENP/Sgo1 fusion constructs to centromeres (as CENP-B fusion, with and without INCENP alpha-helical segment) and by assessing the levels HP1 recruited by these constructs, the authors demonstrate that the combination of PxVxl and alpha-helical segment contributes to robust HP1 recruitment than PxVxl motif alone. Overall, the data presented here convincingly demonstrates the requirement of the additional contribution by the alpha-helical segment downstream of the canonical PxVxl motif for HP1 inner centromere localisation and function.

Major points:

1. According to the ITC data, the INCENP fragments 158-178 and 121-178, spanning the PxVxl motif but lacking the alpha-helical segment, do not show any significant HP1 binding. This is surprising as INCENP PxVxl mutant disrupts HP1 interaction and inner centromere recruitment robustly in cells (Fig 5C and 5C). Possible reasons are not discussed in the manuscript. It is important to show if the INCENP alpha-helical segment alone (in the absence of PxVxl) can interact with HP1 in ITC. Likewise, can INCENP E180A bind HP1 in vitro? If so, what is the measured affinity by ITC? According to Kang et al., 2011, the PxVxl motif of Sgo1 can bind HP1 with ~180 nM affinity. How does INCENP PxVxl motif interaction with HP1 compare with Sgo1 PxVxl motif interaction with HP1? If the mode of interaction is similar, one would expect a similar affinity.
2. The description of 'docking calculation' is not clear, and a detailed description would help. Have the authors tried to generate an AlphaFold-multimer model for INCENP-HP1 interaction and how does the AlphaFold model compare with the model generated using HADDOCK and MODELLER?
3. The authors claim that the disordered region adjacent to the PxVxl motif folds into an alpha helix upon HP1 binding. However, the secondary structure prediction using just the primary sequence of INCENP already suggests the propensity for the amino acids downstream of PxVxl motif to form an alpha-helical structure. This needs to be clarified.

Minor comments:

1. Introduction - where the scaffolding role of INCENP is highlighted, it would be appropriate to include Jeyaprakash et al., Cell 2007 (first crystal structure showing the interaction of INCENP N-terminal helical region forming a helical bundle with Survivin and Borealin).

2. Introduction 2nd paragraph, last sentence - 'These two functional modules are connected by a long intervening unstructured, intrinsically disordered part of INCENP'. Samejima et al., JBC 2015, showed that INCENP has a 32nm long single alpha helix (spanning 213 residues in the central region). Citing this work here will make the introduction more comprehensive.
3. Fig 2B, including the anti-His blot corresponding to the Ni-NTA pull-down, would be helpful.
4. Fig 4D. Measured Kds need to be included for each ITC experiment.
5. Fig 5A. It would be helpful if the experimentally identified INCENP binding surface of HP1 (NMR data) could be highlighted (in a different colour) in this figure. In a way, overlaying what is shown in Figure 4C here.

Reviewer #2 (Comments to the Authors (Required)):

Sako et al demonstrate that INCENP contains additional binding elements C-terminal to its canonical "PxVxI" HP1-binding motif, and provide evidence as to why INCENP is the main recruiter of HP1 at centromeres in mitosis. They propose a model in which one molecule of INCENP binds the symmetric dimer chromoshadow domain of HP1alpha, with a C-terminal helix of INCENP making asymmetric contacts with only one monomer of HP1. Their data suggests that this C-terminal extension is unique amongst HP1 binding proteins.

This paper will be of high interest to people in the field, and I think it is exciting because of its importance to our understanding of CPC regulation, but also HP1 biology in general. However, there are some issues that need to be addressed to ensure that the claims are supported by the data. The two main issues involve the stoichiometry and nature of the binding between INCENP and HP1, and the uniqueness of this mode of binding amongst HP1 binding proteins.

Major Points

- 1) Fig 2D and S2D: Why does the 121-178 INCENP fragment bind in the IP assay but not ITC? Shouldn't the pulldown be more susceptible to off-rates? Why does the 121-178 INCENP fragment have an estimated Kd in Fig. S2D but the binding isotherm in Fig 2D clearly shows no binding? This is very confusing, as many studies have found that the PxVxI motif can bind to HP1 on its own. I think the authors should synthesize a peptide that contains the PxVxI motif and measure binding by ITC, as it is possible that there are solubility issues with the 121-178 fragment. I think this is important to address because their structural data does not include any NOEs, therefore they don't show direct binding to the CSD. Conversely, it would be highly informative to know the affinity of the C-terminal helix for the CSD on its own, as this will help clarify their structural model.
- 2) Are the listed concentrations of HP1 in the ITC and NMR assays in terms of total monomer concentration or of the dimer? Assuming HP1 is 100% dimeric, the dimer concentration should be [monomer]/2. This is important, since the stoichiometry of the interaction is unclear.
- 3) The authors interpret the chemical shift changes in Fig 4 as being indicative of an asymmetric binding event between one INCENP molecule and one of the CSD monomers. However, given their model, it is unclear why the C-terminal portion of a second INCENP molecule can't bind to the symmetric CSD dimer. In this scenario, each molecule of INCENP would be in a dynamic competition with each other for the PxVxI binding site on HP1, but retain the ability to bind using its helical motif (see point 1). So another interpretation of their data is that the blue and red peaks in Fig 4A are representative of different states of exchange. This is where my question in point 2 comes to play. Do you know that you have saturated the binding sites in Fig. 4A? I think the best way to resolve this is to measure the stoichiometry of the complex at different molar ratios by DLS or AUC (the ITC values for N are not accurate, as they are susceptible to deviations due to loss of activity or inaccurate protein concentrations).
- 4) Regarding the model in Fig. 5A, why aren't the authors using AlphaFold or RoseTTA fold? When I put a single molecule of INCENP and a dimer of the CSD into Alphafold multimer, the top hit is a model that is very similar to the one in 5A. However, when I input 2 INCENP molecules or one CSD dimer, it predicts that the each INCENP binds at the same time and that the pocket on HP1 can accommodate both PxVxI motifs. Although I am aware that alphafold can give rise to unrealistic structures, I believe it is truly important to know the actual binding mode of INCENP and HP1.
- 5) Other work has shown that residues C-terminal to the PxVxI motif make asymmetric contacts outside of the main binding cleft (Thiru et al, 2004). Also, the authors only compare INCENP to Sgo1 and Nsl1. I understand why they did that, because they have focused on HP1 in mitosis mainly, but it definitely hampers their claims that the C-terminal extension is unique since they didn't look at other PxVxI-motif containing proteins. Also, is this conserved in INCENP in other species? I would suggest that the authors perform alphafold runs on other known HP1 binding proteins bound to the HP1 CSD and see if there are any other examples of this binding mode to support their claims. Alternatively, they can change the wording to reflect that is potentially

"unique to proteins that bind HP1 in mitosis", however that would alter the impact of the paper in my view.

6) I'm not sure that naming the PVI plus helix the "SSH domain" is appropriate, as it might cause confusion with the SAH domain of INCENP and the acronym doesn't make sense.

Minor Points

1) Abstract, "What secures the specific and predominant interaction is unknown". This won't be clear to many people, unless they are already aware of the previous work from Hongtao Yu.

2) Abstract: please rewrite "these interfaces lie in disordered sequence fold into beta-strand and alpha-helix upon HP1 binding", because it is not clear.

3) Introduction, pg 4: I don't think HP1 can be considered a "fifth subunit of the CPC", because it is not known to be in a stoichiometric complex with other CPC subunits. I think more direct evidence for this would need to be provided (e.g. quantitative sucrose gradients or gel filtration)

4) Intro, pg 4: "In mitosis, most of the HP1 detach [sic] from chromatin...." I think it would be appropriate to cite the Fischle and Hirota Nature 2005 papers here

5) Intro, pg 4: "Unlike INCENP, interaction of Sgo1 with to [sic] HP1 is dispensable for its function in mitosis". I think this misrepresents the findings of the Kang et al paper, where they show that mutating the HP1 binding site on INCENP has no bearing on cohesion, but does cause loss of centromeric HP1. Therefore, I think you should just say "localization" instead of "function".

6) Intro, pg 4: "The fact that INCENP's PVI motif reside within the intrinsically disordered region limits us to obtain structural details of HP1/INCENP interaction". I'm not sure what this means.

7) Fig. 1A: I think the "Vs." nomenclature needs to be explained better or just more simply "INCENP and HP1".

8) Figure 2B: there needs to be MW markers and an explanation of the expected MW of each fragment and what the other bands are. it seems odd that the same amount of HP1 comes down with each fragment, despite clear concentration differences

Response to the reviewers

We are pleased that both reviewers positively evaluated our work and thank them for the thoughtful and constructive suggestions how to further improve the manuscript. We have addressed their specific points as explained below.

Reviewer #1:

Previous studies from the Hongtao Yu (Kang et al., 2011) and Toru Hirota (Abe et al., 2016) labs showed that INCENP possesses an HP1 binding PxVxI motif and INCENP-HP1 interaction mediated by this motif is critical to achieve the full activity of the Aurora B kinase and its effective error-correction role. This manuscript by Sako et al., through detailed structural (NMR and computational modelling) and biochemical characterisation, demonstrates that INCENP-HP1 binding involves not only the PxVxI motif but also a downstream alpha-helical segment. Disrupting the alpha-helix interaction with HP1 (using INCENP E180A mutation) reduced HP1 inner centromere localisation to the same extent as the PxVxI mutant. Consistent with this, the Aurora B activity, as assessed by the Hec1 phosphorylation, was also perturbed by both PxVxI and E180A mutants. Based on these observations, the authors claim that the bipartite mode of INCENP binding to HP1 is crucial for the inner centromere localization of HP1 and the full activity of the CPC. By ectopically tethering different INCENP/Sgo1 fusion constructs to centromeres (as CENP-B fusion, with and without INCENP alpha-helical segment) and by assessing the levels HP1 recruited by these constructs, the authors demonstrate that the combination of PxVxI and alpha-helical segment contributes to robust HP1 recruitment than PxVxI motif alone. Overall, the data presented here convincingly demonstrates the requirement of the additional contribution by the alpha-helical segment downstream of the canonical PxVxI motif for HP1 inner centromere localisation and function.

Major points:

1. According to the ITC data, the INCENP fragments 158-178 and 121-178, spanning the PxVxI motif but lacking the alpha-helical segment, do not show any significant HP1 binding. This is surprising as INCENP PxVxI mutant disrupts HP1 interaction and inner centromere recruitment robustly in cells (Fig 5C and 5C). Possible reasons are not discussed in the manuscript. It is important to show if the INCENP alpha-helical segment alone (in the absence of PxVxI) can interact with HP1 in ITC. Likewise, can INCENP E180A bind HP1 in vitro? If so, what is the measured affinity by ITC? According to Kang et al., 2011, the PxVxI motif of Sgo1 can bind HP1 with ~180 nM affinity. How does INCENP PxVxI motif interaction with HP1 compare with Sgo1 PxVxI motif interaction with HP1? If the mode of interaction is similar, one would expect a similar affinity.

The Reviewer's point is well taken: Given the major role of PVI motif of INCENP in supporting the binding to HP1 in cells, and the PVI motif in Sgo1 can sufficiently support the binding to HP1, it is indeed unexpected that INCENP fragments that contain PVI motif did not interact in ITC unless having the alpha-helical segment together. It is therefore important, we agree, to clarify to which extent do the INCENP alpha-helical segment and the PVI support the binding to HP1 in vitro.

As the reviewer suggested, we first attempted to examine the alpha-helical segment peptide, but unfortunately highly insoluble property of this fragment prevented us from assessing it directly in ITC. Still, the comparison of INCENP 121-270 fragment wild-type (dissociation constant $K_d=0.042 \mu\text{M}$) with its 3A counterpart (K_d : N/A) revealed not only the requirement of PVI motif but also that the alpha-helical segment is insufficient for binding (**Fig. 2 C**). Conversely, comparison of INCENP 121-210 wild-type fragment ($K_d=0.06 \mu\text{M}$) with the fragment bearing E180A mutant ($K_d=0.81 \mu\text{M}$) revealed not only a higher affinity of PVI motif, but also a supporting role of the alpha-helical segment in establishing a stable interaction (**Fig. 6 A; and Fig. S5 A, new right panel**).

Additionally, as one might notice, the ITC measurements were refined for the INCENP 121-178 fragment (**new Fig. S2 E**), which we now purified using the GST system in the revised study (because the purity of previous His-tag version was not sufficiently high, partly due to low expression levels in bacteria). Also, we re-assessed the raw data of the whole ITC assays (**Fig. 2 C and D; Fig. 6 A; Fig. S2 E and F; Fig. S3 E and F; and Fig. S5 A**). This resulted in a slight change in the K_d values, but it was largely reproducible and there were no cases where we needed to reconsider our interpretations.

Thus, the INCENP PVI motif does not support the binding to HP1 by itself, whereas Sgo1 PVI motif does ($K_d=0.18 \mu\text{M}$, Kang *et al.* 2011), if we overlook the difference of HP1 isoforms (former is beta and latter alpha). However, together with the alpha-helical segment, INCENP creates further high affinity to HP1, underscoring the significance of the bipartite binding interface. These points are now unambiguously described in the text as follows:

"We found that INCENP fragments consisting of amino acids 121-270 and 160-210 bound to the HP1 α , whereas the 121-178 fragment and the alanine mutant of PVI motif (Pro167Ala, Val169Ala, and Ile171Ala; PVI_3A) of 121-270 fragment significantly reduced the binding ability (Fig. 2 B). These data indicate that the PVI motif and its downstream sequence are both required to support a stable interaction with HP1.

To verify these results, we conducted isothermal titration calorimetry (ITC) analysis, in which the release or absorption of heat is measured as a readout of molecular interactions. ITC experiments quantitatively showed that INCENP 121-270, 121-210, and 160-210 fragments interacted specifically with HP1 α , whereas 121-178 fragment and a short peptide spanning 159-178 failed to interact with HP1 α despite the presence of the PVI motif (Fig. 2, C and D; and Fig. S2 C-F).” (Results section, page 7)

“Given that the PVI motif, like in many other HP-interacting proteins, plays a major role in INCENP binding to HP1, it was unexpected to find that fragments containing the PVI motif do not bind to HP1 unless they have an α -helical segment (Fig. 2). To what extent do PVI and α -helical segments individually support the binding? The ITC measurements provides insights into this question: Comparison of INCENP 121-270 fragment wild-type (dissociation constant $K_d=0.042 \mu\text{M}$) with its 3A counterpart ($K_d: \text{N/A}$) revealed not only the requirement of PVI motif but also that the α -helical segment is insufficient for binding. Conversely, comparison of INCENP 121-210 wild-type fragment ($K_d=0.06 \mu\text{M}$) with the fragment bearing E180A mutant ($K_d=0.81 \mu\text{M}$) revealed not only a higher affinity of PVI motif, but also a supporting role of the α -helical segment in establishing a stable interaction (Fig. 6 A; and Fig. S5 A). Thus, the INCENP PVI motif dose not bind to HP1 on its own; however, together with the downstream α -helix, INCENP creates high affinity to HP1, underscoring the significance of the bipartite binding interface.” (Discussion section, page 15)

2. The description of 'docking calculation' is not clear, and a detailed description would help. Have the authors tried to generate an AlphaFold-multimer model for INCENP-HP1 interaction and how does the AlphaFold model compare with the model generated using HADDOCK and MODELLER?

According to the comment, we added a detailed description on Docking model in Materials and Methods as follows:

“Docking model

Docking calculations between HP1 α _CSD (111-179) and INCENP (160-192) were carried out using the HADDOCK 2.2 web server (de Vries et al., 2010; Honorato et al., 2021). In these docking calculations, the initial structure of HP1 α _CSD (111-179) was obtained using MODELLER (Martí-Renom et al., 2000) based on the X-ray crystal structure (PDB code: 3I3C), which consists of 111-173 amino acids with missing residues of 132-134. The

initial structure of INCENP (160-192) was derived using the CNS 1.2 (Brünger et al., 1998), incorporating 38 dihedral angles (phi and psi angles for amino acids 165-171, 173, 174, and 177-186) from TALOS analysis (Shen et al., 2009) and six hydrogen bonds between residues 181 and 177, 182 and 178, 183 and 179, 184 and 180, 185 and 181, and 186 and 182 from the main chain chemical shift of the complex, which showed a β -strand (164-171) and an α -helix (177-186). Ambiguous restraints for the docking calculations were classified into two zones based on the complex structure of HP1 β _CSD and Sgo1 (PDB code: 3Q6S), in which a β -sheet was formed between HP1 β _CSD with anti-parallel (168-173) and parallel (168-170) and Sgo1 (448-455) corresponding to HP1 α _CSD with anti-parallel (172-177) and parallel (172-174) and INCENP (164-171). The first zone, where a β -sheet is anticipated to form between HP1 α _CSD with anti-parallel (172-177) and parallel (172-174) and INCENP (164-171), had active residues for HP1 α identified from significant chemical shift perturbations over 0.5 ppm as Ala129, Asp131, Glu169, Arg171, Leu172, His175, Ala176, Tyr177, and for INCENP as Pro167, Val168, Val169, Glu170, Ile171. In the second zone outside the β -sheet formed between HP1 α _CSD with anti-parallel (172-177) and parallel (172-174) and INCENP (164-171), active residues showing significant chemical shift perturbations over 0.3 ppm were identified as Ile113, Arg115, Phe117, Trp142 for HP1 α and Glu180, Gln181, His182, Val183 for INCENP.

In the docking calculations, initially, 1,000 structures were generated using rigid molecular docking. The 200 structures with the lowest energy were first refined by simulated annealing in vacuum, followed by further simulated annealing in a water molecule shell. This refinement of the 200 structures incorporated dihedral angles and hydrogen bond restraints from TALOS analysis. Additionally, distance restraints within 5Å were introduced for N, C', O, CA, and CB of HP1 α _CSD -INCENP corresponding to residues forming β -sheets in the HP1 β _CSD-Sgo1 complex. When clustering was performed on the 200 obtained structures based on the position of the helix (177-186) of INCENP in the complex with a RMSD threshold of 2.5Å, 72 out of the 200 structures were classified into 8 clusters. Among these, the cluster with the most structures and the highest HADDOCK score, consisting of 22 structures, was selected as the final model." **(Materials and Methods)**

In addition, we have carried out intermolecular nuclear Overhauser effect (NOE) experiment, which showed intermolecular NOE signals between one subunit of the CSD dimer labelled by $^{13}\text{C}/^{15}\text{N}$ and nonlabelled INCENP indicating the INCENP asymmetric binding, such that methyl groups from INCENP are close to the amide protons of Ile113, Ala114, and Leu150 from one subunit of the CSD dimer but not from the other subunit (**new Fig. S3 I**).

And our HADDOCK structure explains that the amide protons of Ile113, Ala114, and Leu150 in one subunit are close to the methyl groups of Val183 in the alpha helix of INCENP (**new Fig. S3 J**). On the other hand, AlphaFold prediction indicates that the amide protons of Ile113 of both subunits in the CSD dimer are far from the methyl groups of Val183 of INCENP (**new Fig. S3 K**), which again exemplifies the difference between the observation and the prediction.

To explain these results, we rephrased the text accordingly:

“In addition, we have identified two signals corresponding to intermolecular nuclear Overhauser effect (NOE) between one subunit of the CSD dimer labelled by $^{13}\text{C}/^{15}\text{N}$ and nonlabelled INCENP; amide proton signals of Ile113, Ala114, and Leu150 from one subunit of the CSD dimer showed strong and weak intermolecular NOE signals with methyl groups of INCENP, suggesting the stable complex of the CSD dimer bound to INCENP (Fig. S3 I and J).” (Results section, page 10)

“Although we have not used the NOE results to obtain this docking model (Fig. S3 I and J), it could nevertheless explain that the strong and weak intermolecular NOE signals from amide protons of Ile113, Ala114, and Leu150 of one subunit of the CSD dimer due to two methyl groups of Val183 of INCENP, exemplifying the reliability of the model.” (Figure legends section, page 23)

3. The authors claim that the disordered region adjacent to the PxVxI motif folds into an alpha helix upon HP1 binding. However, the secondary structure prediction using just the primary sequence of INCENP already suggests the propensity for the amino acids downstream of PxVxI motif to form an alpha-helical structure. This needs to be clarified.

This is another insightful suggestion, which we should have made a clear argument of the difference between the observation and the prediction. There are several algorithms predicting the secondary structure from primary sequences, but what they tell is merely the propensity. In light of the INCENP's case where the alpha-helical structure is conditionally formed when interacting to HP1, it is particularly important for the disordered region to recognize that these secondary structure predictions indicate its potential to assemble into a particular secondary structure, and not to make an overestimation. Along this idea, we have added the description accordingly:

“We want to emphasize that, beyond the prediction with AlphaFold algorithm, which predicted the α -helix formation (Fig. S3 K), NMR analysis importantly indicates that it is conditionally assembled from the disordered sequence upon HP1 binding (Fig. 3 D; and Fig. S3 H). Moreover, the exact amino acids linking between the α -helical segment and HP1 α CSD can be identified through the NOE measurements (Fig. S3 I and J). Thus,

predicted structure is useful yet requires experimental verifications.” (Discussion section, page 15)

“(K) The prediction model of AlphaFold2 between INCENP (160-192) and HP1 α CSD (111-191) dimer. Note that the amide protons of Ile113 of both subunits in the CSD dimer are far from the methyl groups of Val183 of INCENP, which exemplifies the difference between the observation and the prediction.” (Figure legends for Supplemental Figures section, page 48)

Minor comments:

1. Introduction - where the scaffolding role of INCENP is highlighted, it would be appropriate to include Jeyaprakash et al., Cell 2007 (first crystal structure showing the interaction of INCENP N-terminal helical region forming a helical bundle with Survivin and Borealin).

We agree with the reviewer and we included the reference as suggested (**Introduction section, page 3**).

2. Introduction 2nd paragraph, last sentence - 'These two functional modules are connected by a long intervening unstructured, intrinsically disordered part of INCENP ...'. Samejima et al., JBC 2015, showed that INCENP has a 32nm long single alpha helix (spanning 213 residues in the central region). Citing this work here will make the introduction more comprehensive.

We agree with the reviewer and we included the reference as suggested (**Introduction section, page 4**).

3. Fig 2B, including the anti-His blot corresponding to the Ni-NTA pull-down, would be helpful.

The reviewer's suggestion is well taken, however anti-His immunoblotting lead us to realize that purity of His-tagged fragments was not sufficiently high. Therefore, we re-setup the experiment using GST-fusion INCENP fragments, which improved the purity and allowed more quantitative analysis. As shown in **new Figure 2 B**, the amounts of GST- INCENP fragments are

now largely equal, both in inputs and in pull-downed samples, and, under those conditions, the amount of bound HP1 convincingly indicates that INCENP 121-178 failed to stably bind to HP1 like the INCENP 121-270 PVI/3A mutant.

4. Fig 4D. Measured Kds need to be included for each ITC experiment.

We agree with the reviewer and we included the measured Kds for each ITC experiment, to improve the visibility (Fig. 2 C and D; Fig. S3 E and F; and Fig. S5 A).

5. Fig 5A. It would be helpful if the experimentally identified INCENP binding surface of HP1 (NMR data) could be highlighted (in a different colour) in this figure. In a way, overlaying what is shown in Figure 4C here.

Following the reviewer's suggestion, we have rearranged the figure (Fig. 5A).

Reviewer #2:

Sako et al demonstrate that INCENP contains additional binding elements C-terminal to its canonical "PxVxI" HP1-binding motif, and provide evidence as to why INCENP is the main recruiter of HP1 at centromeres in mitosis. They propose a model in which one molecule of INCENP binds the symmetric dimer chromoshadow domain of HP1alpha, with a C-terminal helix of INCENP making asymmetric contacts with only one monomer of HP1. Their data suggests that this C-terminal extension is unique amongst HP1 binding proteins.

This paper will be of high interest to people in the field, and I think it is exciting because of its importance to our understanding of CPC regulation, but also HP1 biology in general. However, there are some issues that need to be addressed to ensure that the claims are supported by the data. The two main issues involve the stoichiometry and nature of the binding between INCENP and HP1, and the uniqueness of this mode of binding amongst HP1 binding proteins.

Major Points

1) Fig 2D and S2D: Why does the 121-178 INCENP fragment bind in the IP assay but not ITC? Shouldn't the pulldown be more susceptible to off-rates? Why does the 121-178 INCENP

fragment have an estimated K_d in Fig. S2D but the binding isotherm in Fig 2D clearly shows no binding? This is very confusing, as many studies have found that the PxVxI motif can bind to HP1 on its own. I think the authors should synthesize a peptide that contains the PxVxI motif and measure binding by ITC, as it is possible that there are solubility issues with the 121-178 fragment. I think this is important to address because their structural data does not include any NOEs, therefore they don't show direct binding to the CSD. Conversely, it would be highly informative to know the affinity of the C-terminal helix for the CSD on its own, as this will help clarify their structural model.

The Reviewer correctly pointed out the inconsistent results between the pull-down assay and ITC measurement for the 121-178 INCENP fragment, and its ITC profile indicating no specific binding should not give K_d value. The reason why this fragment failed to behave properly is unknown, but a limited expression in bacteria and an insufficient level of purity might be attributable. Therefore, in the revised study, we purified the INCENP 121-178 fragment using the GST system and repeated these experiments.

In pull-down assays, the amount of HP1 co-precipitated with INCENP 121-178 dropped to the basal level, to a range co-precipitated with INCENP 121-270 PVI/3A mutant (**new Fig. 2 B**). Consistent with these results, the INCENP 121-178 fragment also did not interact with HP1 in ITC assay (Fig. S2 E).

Following the reviewer's suggestion, we synthesized a peptide spanning 134-177 including PVI (as an alternative for the 121-178 peptide, which we failed to synthesize because of the poly Alanine sequence) and used in the ITC analysis as shown below. This INCENP 134-177 peptide showed no binding profile in ITC, indicating that PVI motif in INCENP does not bind to HP1.

Furthermore, the reviewer also asked the affinity of the C-terminal helix for the CSD on its own. The comparison of INCENP 121-270 fragment wild-type ($K_d=0.042 \mu\text{M}$) and the 3A counterpart ($K_d: \text{N/A}$) reveals not only the requirement of PVI motif for binding, but also the C-terminal helix does not sufficiently support the binding by its own (**Fig. 2 C**).

Based on these new experimental datasets, we concluded that, unlike the Sgo1 PVI motif (canonical PVI), the INCENP PVI motif alone does not support the binding to HP1 and requires the downstream helical segment to establish a stable binding to HP1. We accordingly revised the text as follows:

“We found that INCENP fragments consisting of amino acids 121-270, 160-210 bound to the HP1 α , whereas the 121-178 fragment and the alanine mutant of PVI motif (Pro167Ala, Val169Ala, and Ile171Ala) of 121-270 fragment significantly reduced the binding ability (Fig. 2 B). These data indicate that the PVI motif and its downstream sequence are both required to support a stable interaction with HP1. To verify these results, we conducted isothermal titration calorimetry (ITC) analysis, in which the release or absorption of heat is measured as a readout of molecular interactions. ITC experiments quantitatively showed that INCENP 121-270, 121-210, and 160-210 fragments interacted specifically with HP1 α , whereas 121-178 fragment and a short peptide spanning 159-178 failed to interact with HP1 α despite the presence of the PVI motif (Fig. 2, C and D; and Fig. S2 C-F).” (Results section, page 7)

The reviewer also correctly pointed out the NMR analysis falls short of the NOEs data to indicate the direct binding to the CSD. To address this critical point, we have carried out intermolecular nuclear Overhauser effect (NOE) experiment, which showed intermolecular NOE signals between one subunit of the CSD dimer labelled by $^{13}\text{C}/^{15}\text{N}$ and nonlabelled INCENP indicating a stable complex between the CSD dimer and INCENP with asymmetric binding, such that methyl groups from INCENP are close to the amide protons of Ile113, Ala114, and Leu150 from one subunit of the CSD dimer but not from the other subunit (**new Fig. S3 I**).

Of note, our HADDOCK structure prediction recapitulated that the amide protons of Ile113, Ala114, and Leu150 in one of the CSD subunit are close to the methyl groups of Val183 in the alpha helix of INCENP (**new Fig. S3 J**).

We additionally described these details on the complex structure of HP1 α CSD dimer bound to INCENP in main text as follows:

“In addition, we have identified two signals corresponding to intermolecular nuclear Overhauser effect (NOE) between one subunit of the CSD dimer labelled by $^{13}\text{C}/^{15}\text{N}$ and

nonlabelled INCENP; amide proton signals of Ile113, Ala114, and Leu150 from one subunit of the CSD dimer showed strong and weak intermolecular NOE signals with methyl groups of INCENP suggesting the stable complex of the CSD dimer bound to INCENP (Fig. S3 I and J)." **(Results section, page 10)**

"Although we have not used the NOE results to obtain this docking model (Fig. S3 I and J), it could nevertheless explain that the strong and weak intermolecular NOE signals from amide protons of Ile113, Ala114, and Leu150 of one subunit of the CSD dimer due to two methyl groups of Val183 of INCENP, exemplifying the reliability of the model." **(Figure legends, page 23)**

2) Are the listed concentrations of HP1 in the ITC and NMR assays in terms of total monomer concentration or of the dimer? Assuming HP1 is 100% dimeric, the dimer concentration should be [monomer]/2. This is important, since the stoichiometry of the interaction is unclear.

We thank the reviewer for raising this important point. We wrote the concentration of HP1 α in per dimer basis, and now each concentration of HP1 α is clarified in the Figure legends of all NMR experiments. Because it was difficult to attain high concentrated INCENP solutions, and to obtain one-to-one complex solution of the HP1 α dimer by adding INCENP solutions, we concentrated the mixed solution by using Amicon filter. We have added these details as follows:

"In NMR titration experiments, isotopically labeled INCENP proteins ($^{13}\text{C}/^{15}\text{N}$ labeled or ^{15}N labeled) were titrated by gradual addition of unlabeled HP1 α to the NMR tube; the concentrated HP1 α solution was added step by step to the INCENP solution until the free INCENP signals disappear. The INCENP solution was added to isotopically labeled CSD dimer ($^{13}\text{C}/^{15}\text{N}$ labeled or ^{15}N labeled) solution until the free CSD signals disappear and the complex solution was then concentrated for NMR measurements using Amicon Ultra (Millipore). The $^1\text{H} - ^{15}\text{N}$ HSQC spectrum of the INCENP-HP1 α _CSD mixture was acquired at 298 K.

*The intermolecular NOEs between INCENP (160-192) and HP1 α CSD dimer were obtained from 3D $^{13}\text{C}/^{15}\text{N}$ -filtered ^{15}N -edited NOESY (120 ms mixing time) and 3D ^{15}N -edited NOESY (120 ms mixing time) spectra (Breeze, 2000; Schleucher et al., 1994)." **(Materials and Methods sections, page 39-40).***

As mentioned above, the NMR signals of HP1 α CSD dimer by adding INCENP solutions showed doublet signals arising from asymmetric binding of INCENP to HP1 α CSD dimer, without showing any free HP1 α CSD dimer signals. These results indicate that a one-to-one stoichiometry complex of HP1 α CSD dimer bound to INCENP was mostly obtained.

3) The authors interpret the chemical shift changes in Fig 4 as being indicative of an asymmetric binding event between one INCENP molecule and one of the CSD monomers. However, given their model, it is unclear why the C-terminal portion of a second INCENP molecule can't bind to the symmetric CSD dimer. In this scenario, each molecule of INCENP would be in a dynamic competition with each other for the PxVxI binding site on HP1, but retain the ability to bind using its helical motif (see point 1). So another interpretation of their data is that the blue and red peaks in Fig 4A are representative of different states of exchange. This is where my question in point 2 comes to play. Do you know that you have saturated the binding sites in Fig. 4A? I think the best way to resolve this is to measure the stoichiometry of the complex at different molar ratios by DLS or AUC (the ITC values for N are not accurate, as they are susceptible to deviations due to loss of activity or inaccurate protein concentrations).

The reviewer's considerations are well taken, which all comes to the point if the α -helix region alone can support the binding to the CSD. The observations that the INCENP fragment of the alanine mutant of the PVI motif (3A mutant) yet containing the α -helix region significantly reduced the binding ability in both in vitro (**Fig. 2 C**) and in cells (**Fig. 5 B**) indicate that the α -helix region alone does not stably bind to HP1 α CSD.

Remarkably, as replied above for the major comment 1, the intermolecular NOEs of the methyl groups of the INCENP are visible only from one subunit of the CSD dimer but not from the other subunit, which strongly supports the conclusion that INCENP binds asymmetrically.

Fig. 4A (right panel, red) shows the 1H-15N HSQC spectra of 1.3 mM CSD dimer bound to a little excess of 1.6 mM INCENP (160-192). Given the K_d value of 0.04 μ M between INCENP (160-192) and CSD dimer, the binding of the CSD dimer was saturated to one INCENP molecule without showing any free CSD dimer signals. During the titration, we found the signals from the CSD dimer disappeared and new signals from the INCENP-bound CSD emerged. The two signals from the CSD dimer should be originated from two subunit of the CSD dimer asymmetrically bound to one INCENP molecule.

During the NMR titration experiments of INCENP and adding the concentrated CSD dimer, the free and complex signals of INCENP are observed, reflecting a slow exchange binding process, as shown in the attached Figure below (For example, Gly172 signals in free and in complex labeled G172f and G172c can be detected). In Fig. 3 C, the complex NMR spectrum of INCENP bound to CSD dimer without showing any free INCENP signals means that the binding of the CSD dimer to INCENP was saturated at this condition.

4) Regarding the model in Fig. 5A, why aren't the authors using AlphaFold or RoseTTA fold? When I put a single molecule of INCENP and a dimer of the CSD into Alphafold multimer, the top hit is a model that is very similar to the one in 5A. However, when I input 2 INCENP molecules of one CSD dimer, it predicts that the each INCENP binds at the same time and that the pocket on HP1 can accommodate both PxVxI motifs. Although I am aware that alphafold can give rise to unrealistic structures, I believe it is truly important to know the actual binding mode of INCENP and HP1.

We have indeed conducted AlphaFold prediction with INCENP:HP1=1:2, given the stoichiometry as detailed above. As the reviewer pointed out, we could predict a model that is similar but not identical structure we obtained based on NMR analysis (see our response to Reviewer 1 major point 2). We should emphasize that confidence of AlphaFold prediction is not sufficiently strong for INCENP-HP1 binding, as it creates a complex model, with high reliability, for the INCENP PVI/3A mutant and for the HP1 W174A mutant, both of which are known to lack the binding ability to each other (Kang et al., 2011; Abe et al., 2016).

As for the possible 2:2 model, we could still run AlphaFold with 2:2 stoichiometry. The resulting prediction was that INCENP molecules bind to HP1 CSD, accommodating both PVI

motifs simultaneously into the pocket of HP1 dimer; however this model has to rely on less reliable prediction as shown below (Note that red indicate low probability). In addition, previous structural studies demonstrate that a single beta-strand containing PVI-motif is buried in the groove generated between the beta-strands of two CSD (Thiru et al., 2004; Kang et al, 2011; Leopold et al, 2019). With all these pieces of results/information, it is sensible to consider that the formation of 2:2 complex is unlikely.

The rank 5 model which two INCENP molecules bind to HP1 CSD at the same time. Note that this model had quite low reliance because a per-residue model confidence score (pLDDT) of the INCENP fragments were described in "red" suggesting that the configuration of fragments is very low.

5) Other work has shown that residues C-terminal to the PxVxI motif make asymmetric contacts outside of the main binding cleft (Thiru et al, 2004). Also, the authors only compare INCENP to Sgo1 and Nsl1. I understand why they did that, because they have focused on HP1 in mitosis mainly, but it definitely hampers their claims that the C-terminal extension is unique since they didn't look at other PxVxI-motif containing proteins. Also, is this conserved in INCENP in other species? I would suggest that the authors perform alphafold runs on other known HP1 binding proteins bound to the HP1 CSD and see if there are any other examples of this binding mode to support their claims. Alternatively, they can change the wording to reflect that is potentially "unique to proteins that bind HP1 in mitosis", however that would alter the impact of the paper in my view.

We appreciate the reviewer's suggestion to pursue *in silico* search to address the uniqueness and/or generality of the bipartite mode of interaction. To do this, we conducted secondary structure predictions of HP1/INCENP interaction in other species, while the interaction between INCENP and HP1 has only been reported in human and chicken. The secondary structure predictions revealed the conservation of the beta-strand containing PVI or PVI-like sequence and its downstream alpha-helical structure in vertebrates (shown in new Fig. S6 A).

Next, we performed AlphaFold predictions for other known HP1 binding proteins, and found that HP1 interactors can be classified in two groups. One group uses bipartite binding interface (hINCENP and hScc2), while the other canonical binding interface, relying solely on the PVI

motif (hSgo1, hMis14, hCAF1-A, and hTIF1 β (**shown in new Fig. S6 B**). These data would allow us to propose that the bipartite interface is an unconventional binding mode among the HP1 interactors. We have therefore added:

*“To what extent is the HP1-CPC interaction conserved through evolution, and how does its binding mode differ from other HP1 interactors? Based on the secondary structure predictions, the SSH-like domain can be found in vertebrate INCENP homologs (Fig. S6 A). In non-vertebrates, PVI motif become less clear and SSH domain-like structure is no longer evident (data not shown). Despite these predictions, whether the HP1-mediated regulation of the CPC operates in other organisms remain unclear. Capitalizing on AlphaFold2 predictions, we could assume that HP1 interactors can be separated in two groups (Fig. S6 B): One group adopts a bipartite binding interface similar to the SSH domain (e.g., hINCENP and hScc2), and the other uses the canonical binding mode, relying solely on the PVI motif (e.g., hSgo1, hMis14, hCAF1-A, and hTIF1b). These predictions allow us to propose that the bipartite interface is an unconventional binding mode among the HP1 interactors.” (**Discussion section, page 17**)*

6) I'm not sure that naming the PVI plus helix the "SSH domain" is appropriate, as it might cause confusion with the SAH domain of INCENP and the acronym doesn't make sense.

In light of the reviewer's opinion, we have discussed again the name for the domain. We reached to the conclusion that SSH domain would be appropriate, because firstly it is simple, and acronym is a widely used for nomenclature of domains. In particular, following after naming the single alpha-helix domain as SAH domain, SSH will not be too extraordinary to name the other domain of INCENP. They are similar in style (which is not bad) but indeed distinguishable in pronunciation.

Minor Points

1) Abstract, "What secures the specific and predominant interaction is unknown". This won't be clear to many people, unless they are already aware of the previous work from Hongtao Yu.

The reviewer's point is well taken, and we have left out the sentence in question (**Abstract, page 2**).

2) *Abstract: please rewrite "these interfaces lie in disordered sequence fold into beta-strand and alpha-helix upon HP1 binding", because it is not clear.*

The reviewer is correct and we have rephrased the sentence as follows:

“Remarkably, these domains conditionally fold β -strand (PVI motif) and α -helix from disordered sequence upon HP1 binding, and...” (**Abstract, page 2**)

3) *Introduction, pg 4: I don't think HP1 can be considered a "fifth subunit of the CPC", because it is not known to be in a stoichiometric complex with other CPC subunits. I think more direct evidence for this would need to be provided (e.g. quantitative sucrose gradients or gel filtration)*

Following the reviewer's suggestion, we rephrased it as “an accessory subunit of the CPC”, instead of saying it as a fifth subunit of the CPC (**Introduction section, page 4**).

4) *Intro, pg 4: "In mitosis, most of the HP1 detach [sic] from chromatin...." I think it would be appropriate to cite the Fischle and Hirota Nature 2005 papers here*

We thank the reviewer for noting these literatures, both of which are now cited as suggested (**Introduction section, page 4**).

5) *Intro, pg 4: "Unlike INCENP, interaction of Sgo1 with to [sic] HP1 is dispensable for its function in mitosis". I think this misrepresents the findings of the Kang et al paper, where they show that mutating the HP1 binding site on INCENP has no bearing on cohesion, but does cause loss of centromeric HP1. Therefore, I think you should just say "localization" instead of "function".*

Following the reviewer's suggestion, we have rephrased the sentence to avoid any misleading: “Unlike INCENP, interaction of Sgo1 with HP1 is dispensable for its localization in mitosis.” (**Introduction section, page 4**).

6) *Intro, pg 4: "The fact that INCENP's PVI motifs reside within the intrinsically disordered*

region limits us to obtain structural details of HP1/INCENP interaction". I'm not sure what this means.

We have rephrased the sentence as follows:

"The fact that INCENP's PVI motif resides within the intrinsically disordered region limits us to obtain solid details of HP1/INCENP interaction based on crystal structure analysis"

(Introduction section, page 5)

7) *Fig. 1A: I think the "Vs." nomenclature needs to be explained better or just more simply "INCENP and HP1".*

Following the reviewer's suggestion, we now label "INCENP and HP1" instead of using 'vs' in all Fig. 1 A.

8) *Figure 2B: there needs to be MW markers and an explanation of the expected MW of each fragment and what the other bands are. it seems odd that the same amount of HP1 comes down with each fragment, despite clear concentration differences.*

The reviewer raised several technical issues about the pulldown assay in **Fig. 2 B** and we have revised the experiment. Because purity of His-tagged version was not sufficiently high (partly due to low expression levels in bacteria), we re-setup the experiment using GST-fusion INCENP fragments. This improved the purity and allowed quantitative analysis.

As shown in **new Fig. 2 B**, the amounts of GST- INCENP fragments are now largely equal, both in inputs and pull-downed samples, and, under those conditions, the amount of bound HP1 convincingly indicates that INCENP 121-178 have low affinity to HP1 like the INCENP 121-270 PVI/3A mutant. The molecular size markers are indicated (**new Fig. 2 B**).

April 24, 2024

RE: JCB Manuscript #202312021R

Dr. Toru Hirota
Japanese Foundation For Cancer Research
Ariake 3-8-31
Tokyo, Tokyo 135-8550
Japan

Dear Dr. Hirota:

Thank you for submitting your revised manuscript entitled "Bipartite binding interface recruiting HP1 to chromosomal passenger complex at inner centromeres". As you will see, both reviewers are satisfied with the additional data that you provided to address their concerns. Therefore, we would be happy to publish your paper in JCB pending final revisions necessary to meet our formatting guidelines (see details below). In your final version, as requested by the reviewers, you must provide some further explanation in the results and discussion of the meaning of your findings that neither the PVI motif nor the helical segment of INCENP can bind to HP1alpha on their own.

A. MANUSCRIPT ORGANIZATION AND FORMATTING:

- 1) Text limits: Character count for Articles is < 40,000, not including spaces. Count includes abstract, introduction, results, discussion, and acknowledgments. Count does not include title page, figure legends, materials and methods, references, tables, or supplemental legends.
- 2) Figures limits: Articles may have up to 10 main text figures.
- 3) * Figure formatting: Scale bars must be present on all microscopy images, including inset magnifications (you may alternatively indicate the diameter of the inset). Molecular weight or nucleic acid size markers must be included on all gel electrophoresis. To ensure clarity for color blinded readers the use of red/green in images and graphs is discouraged.*
- 4) Statistical analysis: Error bars on graphic representations of numerical data must be clearly described in the figure legend. The number of independent data points (n) represented in a graph must be indicated in the legend. Statistical methods should be explained in full in the materials and methods. For figures presenting pooled data the statistical measure should be defined in the figure legends. Please also be sure to indicate the statistical tests used in each of your experiments (either in the figure legend itself or in a separate methods section) as well as the parameters of the test (for example, if you ran a t-test, please indicate if it was one- or two-sided, etc.). Also, if you used parametric tests, please indicate if the data distribution was tested for normality (and if so, how). If not, you must state something to the effect that "Data distribution was assumed to be normal but this was not formally tested."
- 5) Abstract and title: The abstract should be no longer than 160 words and should communicate the significance of the paper for a general audience. The title should be less than 100 characters including spaces. Make the title concise but accessible to a general readership.
- 6) Materials and methods: Should be comprehensive and not simply reference a previous publication for details on how an experiment was performed. Please provide full descriptions in the text for readers who may not have access to referenced manuscripts.
- 7) * All antibodies, cell lines, animals, and tools used in the manuscript should be described in full, including accession numbers for materials available in a public repository such as the Resource Identification Portal. Please be sure to provide the sequences for all of your primers/oligos and RNAi constructs in the materials and methods. You must also indicate in the methods the source, species, and catalog numbers (where appropriate) for all of your antibodies. Please also indicate the acquisition and quantification methods for immunoblotting/western blots.*
- 8) Microscope image acquisition: The following information must be provided about the acquisition and processing of images:
 - a. Make and model of microscope

- b. Type, magnification, and numerical aperture of the objective lenses
- c. Temperature
- d. Imaging medium
- e. Fluorochromes
- f. Camera make and model
- g. Acquisition software
- h. Any software used for image processing subsequent to data acquisition. Please include details and types of operations involved (e.g., type of deconvolution, 3D reconstitutions, surface or volume rendering, gamma adjustments, etc.).

10) Supplemental materials: There are strict limits on the allowable amount of supplemental data. Articles may have up to 5 supplemental figures, please reduce your count for example by moving some SI data to the main figures and be sure to correct the callouts to reflect any changes. Please also note that tables, like figures, should be provided as individual, editable files. A summary of all supplemental material should appear at the end of the Materials and methods section.

13) ORCID IDs: ORCID IDs are unique identifiers allowing researchers to create a record of their various scholarly contributions in a single place. Please note that ORCID IDs are now *required* for all authors. At resubmission of your final files, please be sure to provide your ORCID ID and those of all co-authors.

Please note that JCB now requires authors to submit Source Data used to generate figures containing gels and Western blots with all revised manuscripts. This Source Data consists of fully uncropped and unprocessed images for each gel/blot displayed in the main and supplemental figures. Since your paper includes cropped gel and/or blot images, please be sure to provide one Source Data file for each figure that contains gels and/or blots along with your revised manuscript files. File names for Source Data figures should be alphanumeric without any spaces or special characters (i.e., SourceDataF#, where F# refers to the associated main figure number or SourceDataFS# for those associated with Supplementary figures). The lanes of the gels/blots should be labeled as they are in the associated figure, the place where cropping was applied should be marked (with a box), and molecular weight/size standards should be labeled wherever possible.

Journal of Cell Biology now requires a data availability statement for all research article submissions. These statements will be published in the article directly above the Acknowledgments. The statement should address all data underlying the research presented in the manuscript. Please visit the JCB instructions for authors for guidelines and examples of statements at (<https://rupress.org/jcb/pages/editorial-policies#data-availability-statement>).

B. FINAL FILES:

-- Cover images: If you have any striking images related to this story, we would be happy to consider them for inclusion on the

journal cover. Submitted images may also be chosen for highlighting on the journal table of contents or JCB homepage carousel. Images should be uploaded as TIFF or EPS files and must be at least 300 dpi resolution.

****It is JCB policy that if requested, original data images must be made available to the editors. Failure to provide original images upon request will result in unavoidable delays in publication. Please ensure that you have access to all original data images prior to final submission.****

****The license to publish form must be signed before your manuscript can be sent to production. A link to the electronic license to publish form will be sent to the corresponding author only. Please take a moment to check your funder requirements before choosing the appropriate license.****

Thank you for your attention to these final processing requirements. Please revise and format the manuscript and upload materials within 7 days. If you need an extension for whatever reason, please let us know and we can work with you to determine a suitable revision period.

Thank you for this interesting contribution, we look forward to publishing your paper in Journal of Cell Biology.

Sincerely,

Karen Oegema, PhD
Monitoring Editor

Andrea L. Marat, PhD
Senior Scientific Editor

Journal of Cell Biology

Reviewer #1 (Comments to the Authors (Required)):

The authors have adequately addressed my concerns both with additional experiments and textual clarifications. However, as the other reviewer noted, the observation that neither PxVxI nor the downstream alpha helix can interact with HP1 with a measurable affinity is rather surprising. Considering this, it would be useful to provide an explicit discussion on this observation with possible speculative reasoning in the discussion section.

Reviewer #2 (Comments to the Authors (Required)):

The authors have satisfactorily addressed the majority of my comments. However, the new data showing that neither the PVI motif nor the helical segment can bind HP1alpha at all are confusing. One would expect some measurable affinity, for each individual segment, given previous work on PVI motifs and binding thermodynamics (ITC can detect Kds of up to millimolar). This would suggest some level of influence of conformational dynamics on the free INCENP protein, and/or on HP1. This possibility should be included and further explored in the results and discussion sections.